# Experimental study on chorus emission in an artificial magnetosphere

Haruhiko Saitoh [1,2] ✉, Masaki Nishiura[1,2], Naoki Kenmochi [2,3] & Zensho Yoshida[1,2]

Wave particle interaction plays an important role in geospace and space weather phenomena. Whistler mode chorus emissions, characterized by non-linear growth and frequency chirping, are common in planetary magneto-spheres. They are regarded as the origin of relativistic acceleration of particles in the radiation belts and pulsating aurora. Intensive theoretical investigations and spacecraft observations have revealed several important features of chorus emissions. However, there is a need to conduct high-resolution and reproducible controlled laboratory experiments to deepen the understanding of space weather. Here, we present the spontaneous excitation of chirping whistler waves in hot-electron high-$\beta$ plasma ($\beta$ is the ratio of the plasma pressure to the magnetic pressure) in an "artificial magnetosphere", a levitated dipole experiment. These experiments suggest that the generation and non-linear growth of coherent chorus emissions are ubiquitous in dipole magnetic configuration. We anticipate that these experiments will accelerate the laboratory investigation of space weather phenomena.

Interactions between waves and particles are important basic processes in a variety of plasma phenomena. Among the many types of waves in plasmas, the right-hand polarized whistler wave couples with electrons over a wide frequency range, comparable to the electron cyclotron frequency $f_{ce}$, and plays important roles in both natural and laboratory plasmas. Active research in this area has offered some important insights into the space weather phenomena[1]. It is generally accepted that whistler mode chorus emission is a major driving mechanism for the relativistic acceleration of charged particles in the geospace and planetary environment[2–7]. Plasmas in magnetospheres often exhibit anisotropy at parallel and perpendicular temperatures, $T_{\parallel}$ and $T_{\perp}$. For example, under the conservation of the first and second adiabatic invariants in a dipole magnetic field, positive ($A = T_{\perp}/T_{\parallel} - 1 > 0$) temperature anisotropy is naturally generated in the strong-field region near the Earth[8]. Subsequent to whistler wave destablization by electron temperature anisotropy with $A > 0$[9,10], a nonlinear process in an inhomogeneous magnetic field may generate chirping chorus emissions that efficiently accelerate particles via wave particle interactions[11–13]. Broadband hiss is another characteristic whistler activity that occurs in the geospace environment[4]. These whistler-related phenomena are widely found in planetary magnetospheres in solar system[14,15], indicating that they are universal phenomena in the dipole field geometry.

Acceleration and transport mechanisms of relativistic particles in the geospace environment by nonlinear interaction with chorus emissions have attracted significant attention and have been intensively studied theoretically and numerically[11–13,16], and through spacecraft observations[17,18]. Laboratory experiments[19–23] provide an opportunity for detailed measurements in a controlled environment, thereby helping the understanding of the whistler wave phenomena in space. A levitated superconducting ring coil creates a dipole magnetic field that is common to planetary magnetospheres.[24–26] Because many important characteristics of whistler and chorus emissions are strongly affected by magnetic geometries, such as those due to instability conditions and selective nonlinear growth of coherent modes, it is advantageous to conduct these experiments in a magnetic geometry that is globally equivalent to the geometry of the planetary magnetosphere.

[1]Graduate School of Frontier Sciences, The University of Tokyo, Kashiwa, Japan. [2]National Institute for Fusion Science, Toki, Japan. [3]Graduate Institute for Advanced Studies, The Graduate University for Advanced Studies, SOKENDAI, Toki, Japan. ✉e-mail: saito@ppl.k.u-tokyo.ac.jp

Owing to the strongly inhomogeneous field strength, the relaxation of the high-$\beta$ plasma in the dipole field creates a peaked pressure profile, as found both in the laboratory and in space. Hasegawa[24] explained the formation of such a structure in real space as a result of a relaxation process in the magnetic coordinate space. A mathematical formulation to describe such self-organization has been obtained based on the idea of foliation in phase space[27]. Recent experiments in levitated dipoles[28–34] demonstrated the remarkable stability of such high-$\beta$ structures, generated by the inward (or up-hill) diffusion of particles into the strong-field region, driven by slow fluctuations[34]. In this study, we focus on the higher-frequency fluctuation activity of Ring Trap-1 (RT-1) and report the spontaneous excitation of the chirping whistler activity of high-$\beta$ plasma in the RT-1 levitated dipole. The "artificial magnetosphere" has realized the laboratory observation of chirping whistler waves in a magnetic geometry that is globally equivalent to the geometry of magnetospheres.

## Results

### Formation of high-beta plasma in RT-1

Experiments were conducted in the RT-1 magnetospheric levitated dipole (see Table 1 for plasma parameters)[25]. As schematically and photographically shown in Fig. 1, a high-temperature superconducting (Bi-2223) dipole field magnet[29] located inside the vacuum chamber was magnetically levitated, using a feedback-controlled lifting electromagnet placed at the top of the chamber. The combination of the two ring coils generates a dipole field configuration with a separatrix (bold line in Fig. 1a) in the confinement region. The magnetic field strength exceeds $B = 0.5$ T near the coil surface and rapidly drops to approximately 5 mT near the outer vacuum chamber wall. In the present experiment, helium plasma was generated by electron cyclotron resonance (ECR) heating with 2.45 GHz microwave at the layer of $B = 87.5$ mT, as plotted with dot lines. The magnetic fluctuations were measured at different poloidal and toroidal positions at the plasma edge, as shown in the figure. The magnetic probe tip consisted of a reverse wound pair of loops with the same area and cable length to measure the magnetic and electric components separately by subtracting and summing the two loop signals. Three-chord 75 GHz (wave length $\lambda = 4$ mm) interferometers at the tangential and vertical ports measure the electron line density. Electrons and ions have a large temperature inequality because of the very low collision frequency of hot electrons with ions ($< \sim 100$ Hz) in the present relatively low-density ($\sim 10^{17}$ m$^{-3}$) conditions. Thus, the thermal pressure of the ECR-heated plasma in RT-1 was mainly due to the hot-electron component. Measurements using X-ray detectors and edge Langmuir probes showed that electrons have multiple temperature components. As shown in Fig.1a, a soft X-ray (SX) image[35] detects an intense hot-electron component in the core confinement region. Measurements with Si(Li) and CdTe detectors confirmed the presence of

high-energy electrons over a wide region within the plasma, including the edge region near the vacuum chamber wall. Pulse height analysis (PHA) of X-ray photons with Si(Li) detectors and CCD camera identified a considerable ratio of hot electrons in the energy range of several keV to a few tens of keV[33]. Interferometry and edge Langmuir probing showed that lower-temperature electrons (around 10 eV) were a major component of the plasma. Further detailed descriptions of RT-1 and the characteristics of ECR-heated plasma are presented in[34] and references therein.

Figure 2 shows the temporal evolution of the plasma formation process and emergence of electromagnetic fluctuations. The plasma was ignited and sustained by the 2.45 GHz ECR heating with an approximately constant power of $P_f = 11$ kW over a duration of 3 s. The electron density remained almost constant throughout the discharge period, except for in the decay phase after $t = 3$ s, as shown in Fig. 2c. The line-averaged electron density was calculated to be $n_{ave} = 1.3 \times 10^{17}$ m$^{-3}$. Because of the temporal evolution of the plasma-wall interactions and the resultant variation in the neutral gas pressure $p_n$, the plasma pressure inferred from the diamagnetic signal $\Delta\Psi$ in Fig. 2d showed rather complex behavior. After the initial rapid ramp up of $\Delta\Psi$ and plasma pressure after t = 0 s, $p_n$ rapidly increased because of the extraction of neutral particles from the chamber wall, which caused the cooling of hot electrons through enhanced neutral collisions, especially at the plasma edge. This is observed as a rapid drop in $\Delta\Psi$ starting at $t = 0.13$ s. Gradual decline of $p_n$ due to the decrease in particle flux from the wall then created a ramp up of plasma pressure through the increase of hot electrons. This phase lasted until the end of microwave injection at $t = 3$ s. Figure 2e shows the increase in electromagnetic fluctuations, which clearly indicates its strong correlation with the plasma pressure sustained by hot electrons. Significant fluctuation activity was observed only for high-$\beta$ states with a considerable ratio of hot electrons ($t = 0 - 0.1$ s and $t = 2 - 3$ s in Fig. 2) and was greatly suppressed under the condition of increased cold-electron component between these two periods.

Through the following experiments, unless otherwise mentioned, we generated plasma in RT-1 under the conditions of $P_f = 11$ kW of 2.45 GHz ECH power injection and an initial filled helium gas pressure of $P_n = 2.5$ mPa. For these experimental conditions, electron density profile was obtained using a global optimization method from measurements with microwave interferometers and an edge Langmuir probe. Figure 3 shows the spatial profiles in the $RZ$ cross-section of RT-1, including the electron density $n_e$, electron plasma frequency $f_{pe}$, electron cyclotron frequency $f_{ce}$, and their ratio $f_{pe}/f_{ce}$. Except for the strong magnetic field and low electron density region in the vicinity of the levitated superconducting coil, $f_{pe}/f_{ce}$ exceeds 1 over a wide region within the plasma, indicating the chorus waves to propagate in most of the plasma interior.

### Fluctuation activities

Figure 4 shows the detailed temporal evolution of each of fluctuation events that were measured with higher sampling rate than that of Fig. 2. As can be also seen from Supplementary Movie 1, the wave activity consists of the abrupt emergence of discrete fluctuation events with a typical duration of 10 μs that continuously appear in the hot-electron high-$\beta$ state of plasmas. These activities classified into three categories, two of which are electromagnetic waves and one is an electrostatic wave. The most active mode appears in the frequency range of $f = 15 - 55$ MHz (approximately $0.1 - 0.4 f_{ce}$). As explained in the Methods section (see below), plasma has a hot-electron component with approximately 9 keV in the present formation conditions. In the high-$\beta$ plasma generated by ECR heating, electrons have temperature anisotropy, especially near the equator of the dipole field. When we assume that the upper limit of this most active frequency range is set by the condition for an unstable whistler frequency in the

**Table 1 | Comparison of typical plasma parameters in the RT-1 experiment and the geospace environment**

|  | RT-1 | Geospace |
|---|---|---|
| $n_e$ | $10^{15} - 10^{17}$ m$^{-3}$ | $10^6 - 10^9$ m$^{-3}$ |
| $B$ | $5 \times 10^{-3} - 5 \times 10^{-1}$ T | $10^{-7} - 10^{-5}$ T |
| $f_{pe}$ | 500 MHz – 4 GHz | 9 kHz – 300 kHz |
| $f_{ce}$ | 140 MHz – 20 GHz | 3 kHz – 300 kHz |
| $f_{pe}/f_{ce}$ | 0.1–10 | 1–10 |
| $T_e$ | thermal, hot (~10 keV) | thermal, warm, hot |

Parameters include the electron density $n_e$, magnetic field strength $B$, plasma frequency $f_{pe}$, electron cyclotron frequency $f_{ce}$, and electron temperature $T_e$.

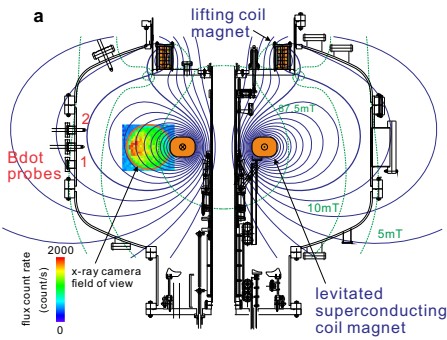

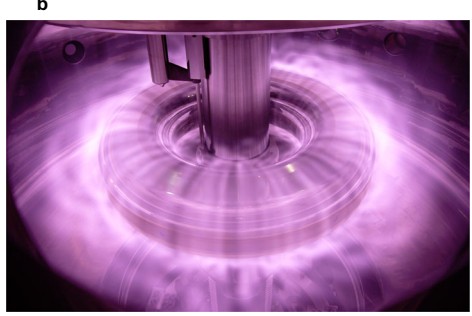

**Fig. 1 | Experimental setup and plasma formation in RT-1. a** Inside the vacuum chamber, a high-temperature superconducting magnet, with current radius 25 cm and mass 110 kg, is levitated by a feedback-controlled lifting magnet coil located at the top of the chamber. The combination of these two coils creates an artificial magnetosphere whose magnetic surfaces and field strength are plotted as thin lines. Plasma is generated by injecting 2.45 GHz microwave and ionizing filled

helium gas. Soft X-ray image in the relatively strong-field region, shown as a color contour map of the flux count of incident photons, indicates the existence of considerable ratio of hot-electron component in the plasma. **b** Visible-light picture of plasma confined around the levitated superconducting coil. The electro-magnetic fluctuations of plasmas were measured with Bdot probes located at different toroidal and poloidal positions.

linear theory[9,10] of

$$f < f_{ce} A/(A+1), \qquad (1)$$

the electron temperature anisotropy is estimated to be $A = T_\perp/T_\parallel - 1 = 0.7$, which is fairly consistent with Grad-Shafranov equilibrium analysis including the temperature anisotropy[36–38].

Waves in this frequency range are usually strongly coherent and exhibit diverse temporal evolutions. As shown in Fig. 4b, waves exhibit down-frequency chirping, up-frequency chirping, and sometimes a combination of the repetition of fast down and up chirping with slower frequency variation, together with continuous growth and decay of the fluctuation amplitude. Interestingly, the simultaneous emergence and intersection of separate fluctuation events with different frequencies and chirping rates were also observed, as shown in Fig. 4b. A comparison of the measurements with a pair of reversely wound magnetic loops indicates that fluctuations in this frequency range are generally electromagnetic waves. Other coherent electromagnetic events were also found in the frequency range of 70−100 MHz (approximately $0.5 − 0.7f_{ce}$) with a lower occurrence rate. These are typically weak down-chirping events with a lower appearance rate as shown in Fig. 4a. Above this frequency range and below $f_{ce} = 140$ MHz, there also appear rather broadband, mainly electrostatic oscillations.

As indicated in Fig. 4, fluctuations are primarily detected in the region below around 120 MHz, but are rarely observed beyond 150 MHz. As shown in Fig. 3, on the other hand, the electron cyclotron frequency exceeds 10 GHz in the stronger field region of RT-1. If the whistler emission occurred in such a stronger magnetic field region, transmitted, and detected at Bdot probes at the plasma edge, it is highly unnatural that there is an almost complete absence of detectable fluctuation activities above 120 MHz (~1% of the electron cyclotron frequency $f_{ce}$ of the strong-field region). Based on these observations, we anticipate that the source of fluctuations is localized in the weak magnetic field region at the edge plasma.

**Dependence on density and plasma pressure**

In Fig. 5, multi-point magnetic diagnostics show the toroidally localized nature of the observed fluctuations. Fluctuations were simultaneously measured at the poloidally and toroidally separated locations at positions at 1, 2, and 3, as shown in Fig. 5a. Locations 1 and 2 were at the north (N) ports in the same poloidal cross-section of the device on the same field line. Location 3 was at the southwest (SW) port of RT-1, toroidally 135° from the north ports. As shown in the figure, for most of the events (more than ~80% of the total events) the onset of instability

was synchronized on probes 1 and 2, whereas probe 3 measured fluctuation events independently of probes 1 and 2. With measurements on the same poloidal cross-section in Fig. 5c, we detected the propagation of fluctuation wave along magnetic field lines. These characteristics are in marked contrast to the previously reported hot-electron-induced interchange (HEI) instability[39,40], which has a flute structure that rotates in the toroidal direction. The parallel wave numbers inferred from measurements with separately placed Bdot probe with a vertical distance of 1 cm along filed line are plotted for waves with different frequencies, as shown in Fig. 5d. The dispersion relations of the whistler mode calculated for the magnetic field strength and electron density at the fluctuation measurement position of plasma edge, $B = 5.4$ mT and $n_e = 6.3 \times 10^{14}$ m⁻³ is shown as a solid line in Fig. 4d, which shows relatively good agreement with the measurement results. Thus, we conclude that the spontaneously excited mode is a whistler wave initially destabilized by temperature anisotropy in the ECR-heated plasma. Despite being whistler waves caused by the temperature anisotropy of high-energy electrons, it is noteworthy that the wavelengths observed differ significantly due to differences in the equipment and environment[19–23]. Further studies are needed for the comparison of the wave excitation and propagation properties with other experiments.

Experimental conditions in which the whistler fluctuation activity is spontaneously excited in RT-1 are shown in Fig. 6 for various electron densities and diamagnetic signal $\Delta\Psi$ (plasma pressure sustained by electrons) values. A threshold value of $\Delta\Psi$ exists to excite the active whistler waves. In the coil levitation operation, a clear whistler activity was observed above $\Delta\Psi \sim 1.2$ mWb. According to equilibrium analysis[36], the maximum local $\beta$ value was 22% for this case. Stabilization of the whistler waves is also realized by decreasing the ratio of hot electrons by increasing the bulk thermal electrons (higher density region above $n_e \sim 1.2 \times 10^{17}$ m⁻³ in Fig. 6). The upper limit of both the appearance frequency and fluctuation amplitude is set by the incident power of the 2.45 GHz microwave in the present conditions. These observations indicate that hot electrons with temperature anisotropy in the ECR-heated plasma are the free energy that destabilize the fluctuations.

In this work, spontaneously excited chirping electromagnetic and rather broadband electrostatic waves were experimentally investigated in the RT-1 levitated dipole. Chorus emission has been shown to appear ubiquitously in laboratory and space environments for various parameters when plasma satisfies certain conditions. Investigations conducted in a controlled laboratory environment with higher reproducibility and high-resolution diagnostics are expected to provide a

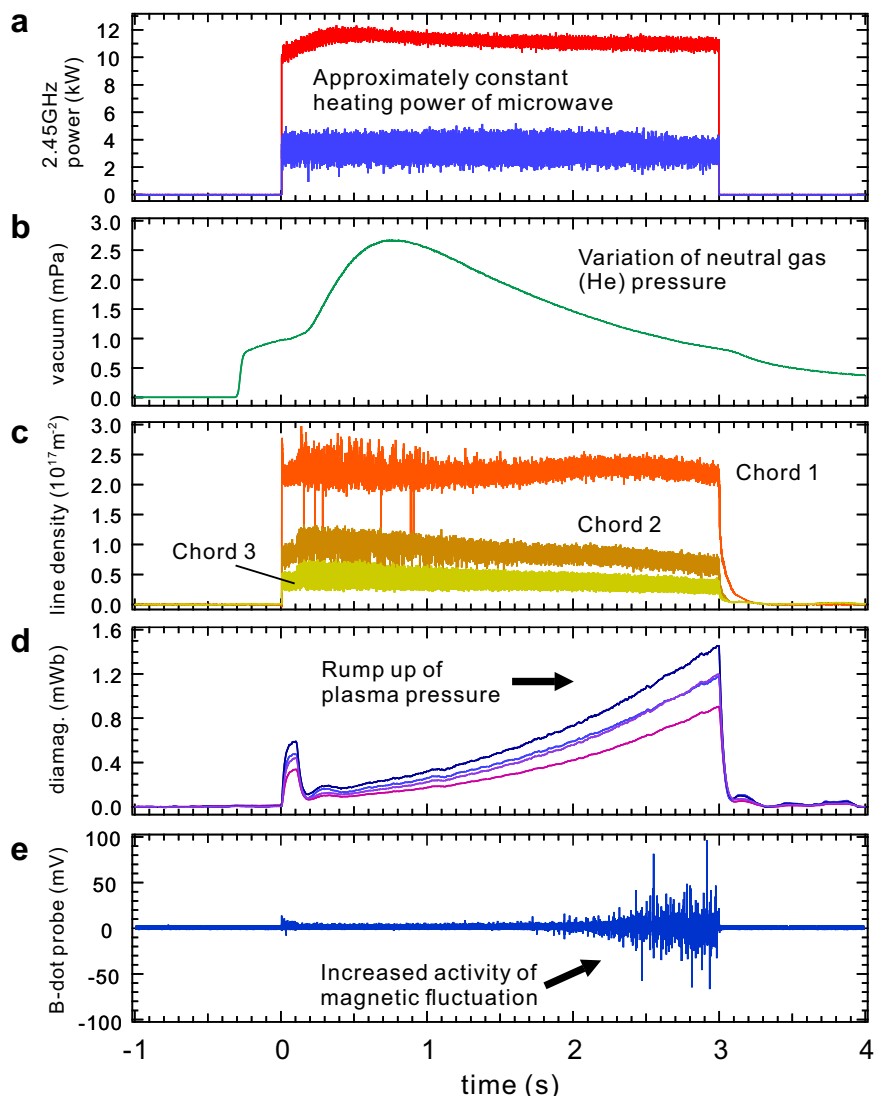

**Fig. 2 | Emergence of electromagnetic fluctuations in hot-electron plasma.** Time traces of plasma formation, including **a** incident and reflected microwave power $P_f$ and $P_r$, **b** filled helium gas pressure, **c** line-averaged density $n_{ave}$ measured with three-chord 75 GHz interferometry, **d** diamagnetic signals $\Delta\Psi$ reflecting the plasma pressure at four different positions, and **e** magnetic fluctuation activity measured at the edge confinement region. The plasma fluctuation in Fig. 2e is active only when the plasma has a large thermal pressure caused by the hot-electron component, as seen in Fig. 2c.

complementary approach to further understanding the chorus emission phenomena that are common in both laboratory and geospace plasmas. In the future, this research might contribute to the understanding of wave-particle interactions in the space weather phenomena, which has a critical impact on advanced technologies[1].

## Methods

### The RT-1 experiment

This study was conducted at the RT-1 levitated dipole experiment[29]. The origin of plasma experiments in internal coil devices dates back to the 1970s, when the magnetohydrodynamic (MHD) properties of high-temperature plasmas were intensively studied in spherator and levitron devices[41]. This type of experiment have attracted renewed interest since the 1990s, following the discovery of extremely high-$\beta$ flowing plasma in planetary magnetospheres and the progress of relaxation theories[32]. Using a superconducting (SC) coil with a persistent current as a current ring, it is possible to mimic the magnetospheric environment in a laboratory with minimal perturbation to plasmas. Recent levitated dipole experiments include RT-1[25] at the University of Tokyo, LDX[26] at MIT and Columbia University, and APEX-D[42] at the Max Planck

Institute for Plasma Physics for the creation of magnetically-confined electron-positron plasmas.

The levitated coil of RT-1 is a 2160 turn winding made of Bi-2223 high-temperature SC tape. The coil is usually cooled from room temperature to below 20 K over two days using a helium gas circulation system with three 10 K cryocoolers. After cooling, the coil is directly charged to the rated current value of 250 kAT with an external power supply using a persistent current switch (PCS) made of a thin YBCO film. During the levitation operation, the heat input into the SC coil is maintained below 1 W, which is realized by the thermal insulation structure installed between the SC coil and a stainless-steel coil case. This enabled more than six hours of plasma experiments before the coil temperature reaches the operational limit of 30 K.

The key to realizing a magnetospheric environment in a laboratory is the stable magnetic levitation of the SC dipole field coil. Here, we briefly explain the method used to levitate an SC coil to create an artificial magnetosphere in a laboratory. The geometry of an artificial magnetosphere is generated by a coaxially located pair of ring coils, namely, a floating superconducting (SC) coil (F coil) in a vacuum chamber and a lifting coil (L coil), usually located at the top of the

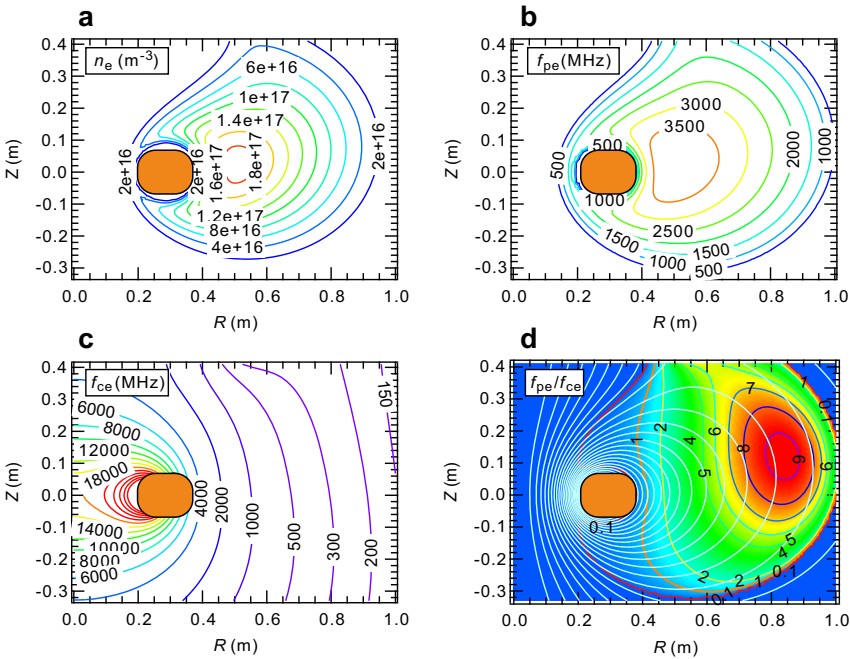

**Fig. 3 | Several plasma parameters in the _RZ_ cross-section of RT-1. a** electron density $n_e$, **b** plasma frequency $f_{pe}$ **c**, electron cyclotron frequency $f_{ce}$, and **d**, their ratio $f_{pe}/f_{ce}$. Pale blue thin lines in **d** shows the magnetic surfaces generated by the levitated superconducting coil and lifting coil.

chamber. The F coils are cooled to cryogenic temperatures below the critical temperature of the SC and is levitated in the vacuum chamber. The coil motion in this system is determined by two basic equations, namely the equation of motion for the F coil and the magnetic flux conservation equation of an SC loop (the so-called fluxoid conservation). The latter holds true because the total flux of an SC F coil is conserved, independent of the variation in the F coil current. These relations determine the behavior of the F coil, such as the stability of motions and the typical oscillation frequency.

One may define the growth rate α of the vertical magnetic force $F_{mz}(z)$ on the coil where $\alpha(z) = (dF_{mz}/dz)/F_{mz}$ and $z$ is the vertical coil position. The vertical coil motion is unstable when $\alpha$ is positive; when the F coil moves upward toward the L coil, the F coil feels an increasing attractive force, and vice versa. Although numerical calculations may be needed for precise stability analysis of vertical coil motion in general, we can readily imagine that α is often positive, at least when ignoring the flux conservation effects. This assumption implies that the current or the magnetic field strength of the F and L coil magnets is constant. An example of such a system us the use of a pair of permanent magnets. When an attempt is made to levitate one magnet using an attraction force from the other magnet under gravity, it can be immediately noticed that the magnet motion shows an unstable positive feedback, indicating that an external stabilizing mechanism may be needed in the levitated dipole experiment.

By solving the aforementioned two governing equations for the F coil motion, two equilibrium current sets are obtained as solutions for one F coil position, provided that the F coil is not too heavy and can be levitated by the L coil field strength. The first equilibrium is relatively weakly affected by the flux conservation law of the F coil. Most of (or presumably all of) levitated dipole experiments, such as LDX, RT-1, Mini-RT, use this equilibrium configuration. This is quite often well approximated by an equilibrium completely ignoring the flux conservation effects. Because the SC F coil current is slightly reduced after turning on the L coil, the levitation force is compensated for by a slightly increased L coil current. As previously explained, this equilibrium is unstable for vertical motion and requires a feedback-control

system. In the other equilibrium position, the persistent current of the F coil is significantly reduced. Levitation is then realized by a very large L coil current, which mainly sustains the magnetic flux inside the F coil loop. This equilibrium is often very stable vertically, which means that the F coil position can be fixed without any external control system. However, in many plasma experiments, such equilibrium is generally not realistic. This is because of the very large L coil current and the resultant small confinement volume of plasmas.

Because coil motions are three-dimensional, considerations of other modes are also needed in addition to the aforementioned vertical motion. The F coil motion can be classified into three categories, vertical, slide, and tilt. The coil behaviors strongly depend on the relative positions of the F and L coils. According to the linear analysis of coil motion stabilities, vertical and slide motions are alternatives; we cannot stabilize or destabilize both of them simultaneously. Tilt instability must also be avoided by selecting an appropriate coil configuration. Because the slide and tilt motions are two-dimensional, it is clear that the detection and stabilization of these modes are not straightforward. In contrast, vertical motion is greatly advantageous since it is one-dimensional. Therefore, it is possible to operate the F and L coils so that the system is spontaneously stable for the slide and tilt motion, and to stabilize the unstable vertical motion using an external feedback-control system.

## Electron density profile

For the measurement of hot-electron high-$\beta$ plasma created by electron cyclotron resonance heating (ECH) with 2.45 GHz microwaves, the diagnostics system of RT-1 includes interferometry to measure line-integrated electron density, diamagnetic loops to measure the stored plasma energy, X-ray detectors to diagnose hot electrons in plasma, and magnetic Bdot probes to measure electromagnetic fluctuations.

Three sets of $f = 2\pi\omega = 75$ GHz microwave interferometers were used to measure the line-integrated electron density in the RT-1 experiment. They were located at the tangential ports (chord1) and vertical ports (chords 2 and 3) of the vacuum chamber. For a diagnostic

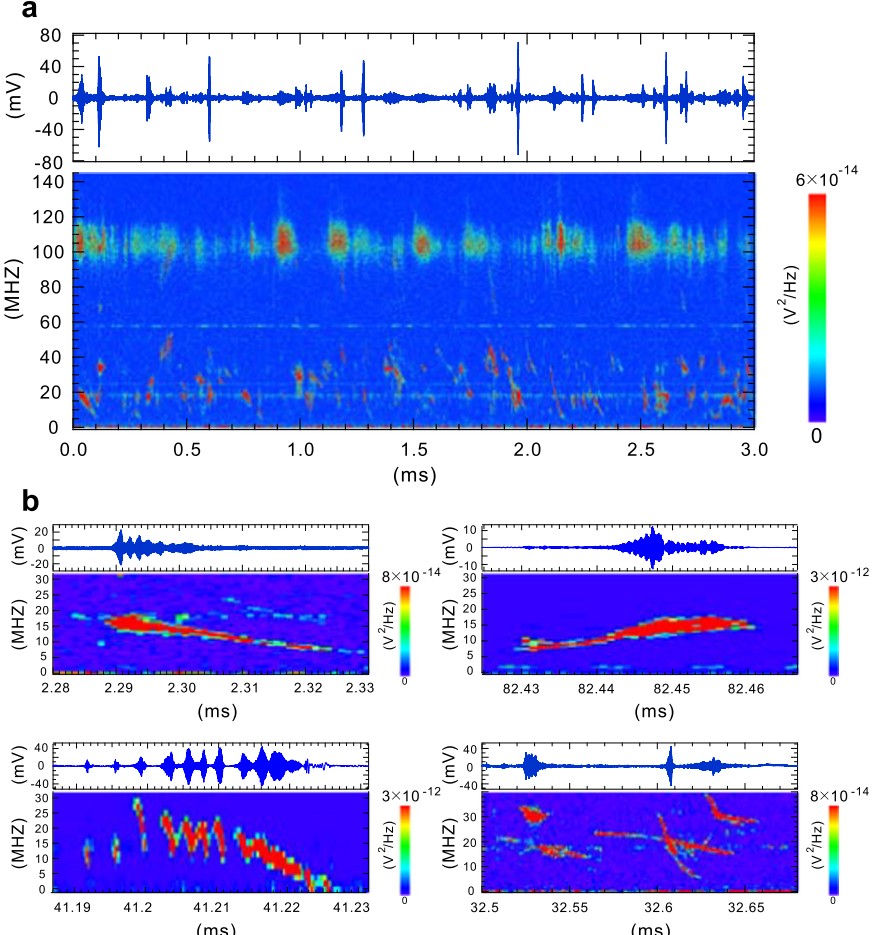

**Fig. 4 | Temporal evolution of plasma fluctuation activity.** Each of the frames shows the time trace of the fluctuation signal and its frequency power spectrum. **a** Temporal evolution of electromagnetic and electrostatic fluctuations in the hot-electron high-$\beta$ mode of RT-1. The fluctuation activities consist of abrupt and repeated discrete fluctuation events. **b** The temporal evolutions of typical fluctuation events measured with same plasma formation conditions but different plasma discharge shots. These events are classified into down chirping, up chirping, multiple repeat of down and up chirping, and simultaneous appearance of different wave events. These complicated temporal evolutions of frequency indicate the nonlinear interaction between fluctuation waves and plasma particles. Plasma was generated with $P_f = 11$ kW of 2.45 GHz ECH power and an initial filled helium gas pressure of $P_n = 2.5$ mPa.

microwave that satisfies the condition

$$\omega_{pe}{}^2, \omega_{ce}{}^2 \ll \omega^2, \tag{2}$$

where $\omega_{pe} = \left(n_e e^2 / \varepsilon_0 m_e\right)^{1/2}$ is the plasma frequency and $\omega_{ce} = eB/m_e$ is the electron cyclotron frequency, a reflective index in the plasma with finite electron temperature is approximated as

$$N = 1 - \gamma X/2 \tag{3}$$

independent of the propagation direction of the wave against the magnetic field. Here $\gamma = 1 - 5/2\mu$ and $\mu = m_e c^2 / k_B T_e$. Because the phase shift observed by an interferometer is given by

$$\Delta\phi = \frac{2\pi}{\lambda} \int (1 - N)\, dl = \frac{\pi}{\lambda} \int (\gamma X)\, dl, \tag{4}$$

the line-integrated electron density that corresponds to a phase shift of $\Delta\phi = 2\pi$ is

$$\int n_e\, dl = 7.5 \times 10^6 \times f/\gamma. \tag{5}$$

When the relativistic effect of hot electrons is neglected, the phase shift of $2\pi$ gives the line-integrated density of

$$\int n_e\, dl = 5.6 \times 10^{17}\, \mathrm{m}^{-2} \tag{6}$$

In order to assess information regarding the transmission conditions and sources of whistler wave propagation throughout the entire plasma, we performed a density distribution reconstruction using a global optimization technique. To incorporate information from the plasma edge region where fluctuation measurements were taken, we included information from the edge Langmuir probes in the evaluation function. We then used the difference between the computed density distribution and the experimental data, represented with multiple parameters, to search for a density profile solution with the minimum of an evaluation function value in the entire computational domain.

Once the search for a single solution converged to a local minimum, we utilized an algorithm that searches for local minima again from initial values that are randomly displaced from the previous one in both direction and distance. This approach allows us to explore local minima across the entire computational region. Figure 7a illustrates the convergence of the global optimization algorithm for a test

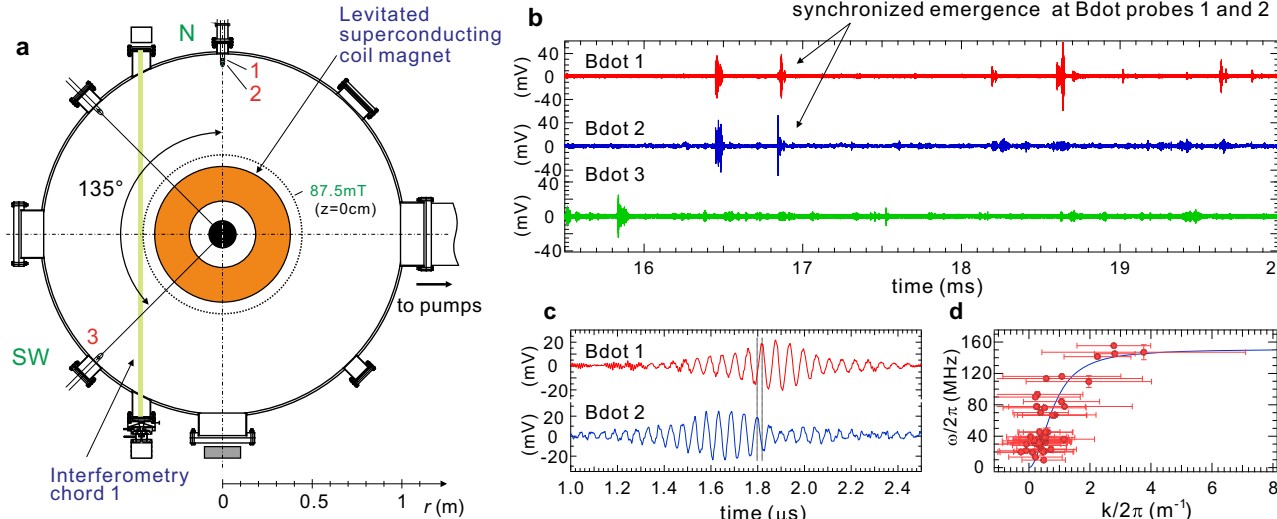

Fig. 5 | Spatial structure of fluctuations measured at different positions. a The top view of RT-1 experiment. Bdot probe 1 ($z = 0$ cm) and 2 ($z = +12$ cm) were located at the north (N) port and Bdot probe 3 ($z = 0$ cm) was at the southwest (SW) port. b Magnetic fluctuation signals simultaneously measured at three different positions indicate that the fluctuation event is toroidally localized. c Comparison of waveforms on the same field line of the dipole field. d Dispersion relation for local electron density and magnetic field strength at the Bdot probe position (line) and

measured values of wave number and frequency (circles). Each fluctuation event data was divided into segments of equal length, specifically 1.6384 μs with 4096 data points. Within each segment, averaged phase difference was computed to obtain wave number information. The resulting wave number values for each event are plotted with their mean and the standard deviation as error bars. Plasma was generated with $P_f = 11$ kW of 2.45 GHz ECH power and an initial filled helium gas pressure of $P_n = 2.5$ mPa.

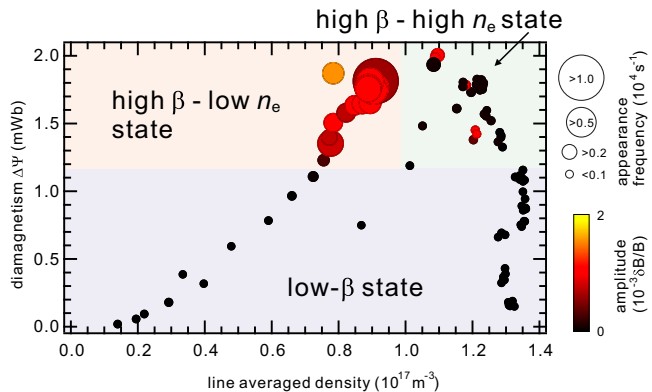

Fig. 6 | Dependence of fluctuation activities on plasma parameters. Occurrence rate (circle size) and amplitude (color contour) of whistler fluctuations in the parameter space of RT-1 for 1 s discharge. The 2.45 GHz ECH power $P_f$ was between 1 and 19 kW, and the initial filled helium gas pressure $P_n$ was between 1.5 and 7.5 mPa. The fluctuations are active in high-$\beta$ hot-electron plasmas, and are suppressed by decreasing the plasma pressure (low-$\beta$ state) or by increasing the cold-electron component (high-$\beta$ - high $n_e$ state).

evaluation function $F$ set for a case with two variables $x_1$ and $x_2$, successfully detecting all the local minimum values. From the detected local minimum values, we selected the most suitable one as the solution.

In order to accurately reflect information from the plasma edge where oscillations were measured, we conducted a search for solutions using measured interferometer data, $I_{mi}$, along with electron density measurements, $n_m$, obtained using edge probes near the vacuum vessel walls. Initially, on the equatorial plane of the device, we provided density in the region $\psi_1 \leq \psi \leq \psi_2$ between the coil surface (magnetic flux function $\psi = \psi_1$) and the vacuum vessel wall ($\psi = \psi_2$) as:

$$n(r, z = 0) = n_0 |\psi - \psi_1| |\psi - \psi_2|^a \qquad (7)$$

As illustrated in Fig. 7b, by choosing the values of parameters $n_0$ and $a$, it is possible to select the absolute value of density and the shape of the density distribution within the confinement region with high flexibility.

On the same magnetic field line, we accounted for the dependence on the magnetic field by calculating:

$$n(r, z) = n(r, z = 0) \times \left(\frac{B}{B_0}\right)^b \qquad (8)$$

where $B_0 = B(r, z = 0)$ is the field strength on the equator and $B(r, z)$ is that of each of the measurement point. This method allowed us to incorporate the dependence on the magnetic field into the density distribution on the $RZ$ plane. We then computed quantities that can be obtained through measurements. These include the line-integrated density in the line of sight as measured by the microwave interferometer:

$$I_{ci} = \int n(r, z) dl_i \qquad (9)$$

and the local density measured at the same radial position as the Bdot probe with the Edge Langmuir probe:

$$n_{ce} = n(r = 0.98, z = 0) \qquad (10)$$

With these quantities, we reconstructed the density by searching for parameters that minimize the function:

$$F = \sum_i (I_{ci} - I_{mi})^2 + (n_{ci} - n_{mi})^2 \qquad (11)$$

We conducted the search for parameters with experimental measurements of interferometer results: $I_{m1} = 2.234 \times 10^{17}$ m$^{-2}$, $I_{m2} = 7.056 \times 10^{16}$ m$^{-2}$, $I_{m1} = 3.203 \times 10^{16}$ m$^{-2}$, and edge density $n_{r=0.98} = 3.012 \times 10^{15}$ m$^{-3}$. The search for solutions resulted in the best-fit parameters: $n_0 = 2.347 \times 10^{11}$, $a = 1.082$, and $b = 0.6375$. The electron

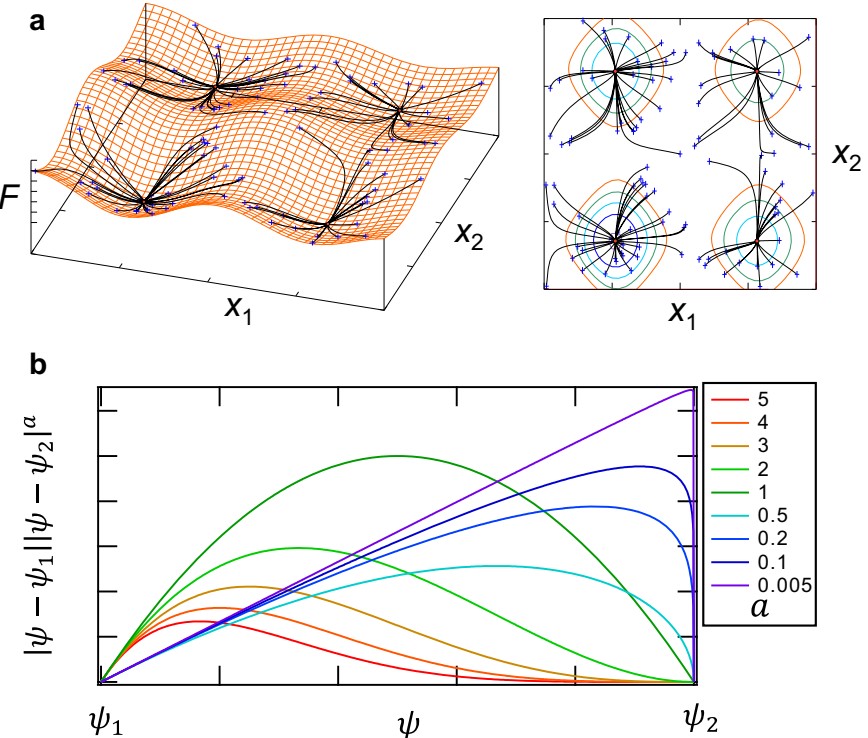

**Fig. 7 | Example of search of solution in the technique of a global optimization method and Shape of electron density profile on the equator of the device. a**, Starting from randomly distributed initial values (blue crosses), all local minimum values (red circles) are detected in the entire computational region. **b**, By changing $a$, various shapes of density profiles are represented.

density $n_e$ and the corresponding electron plasma frequency $f_{pe}$ distributions in the $RZ$ section of the RT-1 device are shown, along with the electron cyclotron frequency $f_{ce}$ with respect to the magnetic field configuration ($f_{pe}/f_{ce}$).

## Diamagnetic measurement and plasma pressure

Diamagnetic signals were measured using four magnetic loops wound at the outer surface of the vacuum chamber at $r = 101$ cm. The equilibrium pressure of the plasma was reconstructed using Grad-Shafranov numerical analysis based on these magnetic measurements. The Grad-Shafranov equation provides magnetohydrodynamic equilibria for axisymmetric plasmas, and solving it numerically allows us to obtain the equilibrium distribution of plasma[36–38]. According to the numerical analysis, the averaged diamagnetic signal $\Delta\Psi$ and maximum local $\beta$ value approximately satisfy $\beta$ (%) $\sim 18 \times \Delta\Psi$ (mWb). The plasma pressure was mainly responsible for the hot electrons under the present experimental conditions.

## Electron temperature

The ECH-generated plasma of RT-1 has several electron populations at different temperatures. The bremsstrahlung from hot electrons was detected using Si(Li) and CdTe detectors, and an X-ray CCD camera. Edge Langmuir probes were used to measure cold electrons in low-density peripheral regions. These cold electrons are generated primarily by collisions with high-energy electrons and ionization of neutrals, and have temperatures of around 10 eV. The ratio of the two populations was controlled by varying plasma production conditions. In RT-1, the hot-electron component have been measured with the Pulse height analysis (PHA) method using Si(Li) and CdTe detectors. In the operation conditions presented in this work, ECH microwave power of 11 kW and filling helium gas pressure of 2.5 mPa, we measured the temperature of the high-energy electron component, primarily responsible for plasma pressure, using a Si(Li) detector. When

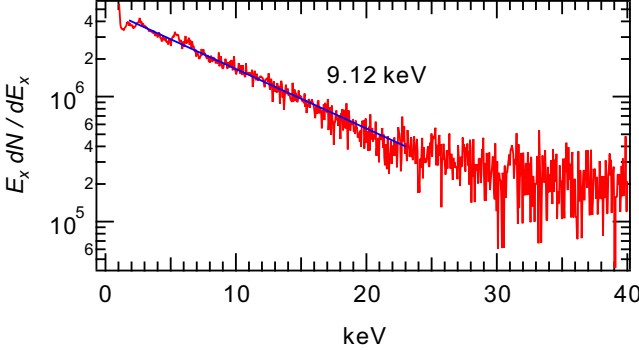

**Fig. 8 | X-ray energy spectrum emitted from the hot electrons of RT-1 plasma.** Electrons consist of bulk cold component and energetic components.

electrons follow a Maxwellian distribution with a temperature, $T_e$, the continuous spectrum emitted from the plasma can be described in terms of the number of photons per unit energy as

$$\frac{dN}{dE_x} = \alpha n_e n_i \frac{g_f \exp(-E_x/T_e)}{E_x \sqrt{T_e}} \tag{12}$$

where $g_f$ is the Gaunt factor, $n_e$ is electron density, $n_i$ is ion density, and $\alpha$ is a constant. Figure 8 represents the energy spectrum of X-rays obtained in this manner, and the high-energy electrons responsible for plasma pressure are estimated to have a temperature of approximately 9 keV. The relatively good linearity of X-ray intensity with photon energy suggests that these high-energy electrons have a distribution rather close to a Maxwellian distribution.

The spatial profile of hot electrons was measured with an X-ray pinhole camera with a CCD of 1024 × 1024 pixels and a 16-bit dynamic

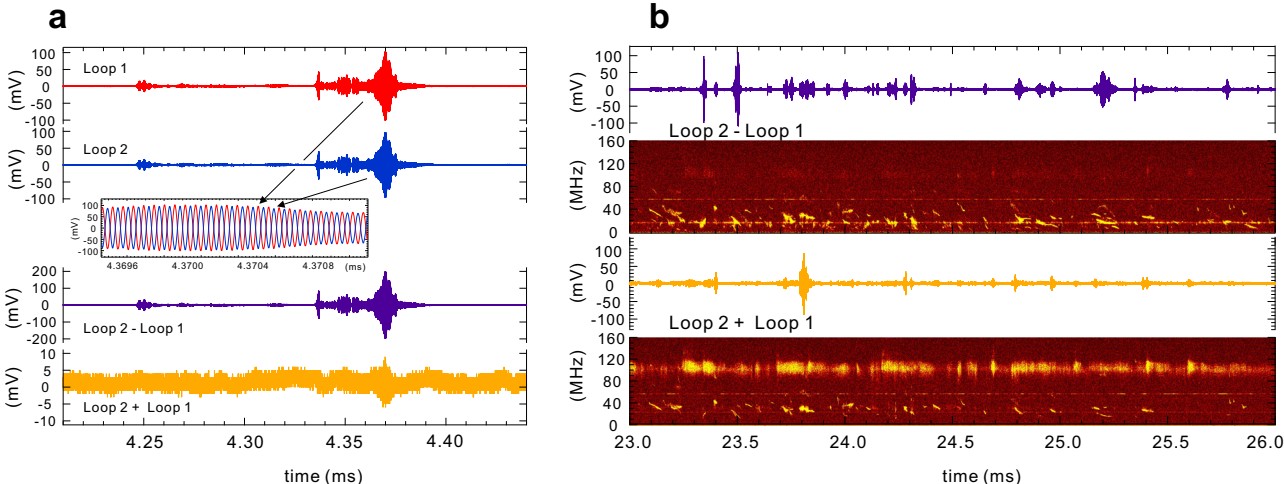

**Fig. 9 | Temporal evolution of a typical electromagnetic fluctuation event and separation of magnetic and electrostatic fluctuation signals. a** By adding or subtracting the signals of loops 1 and 2, magnetic and electrostatic signals are detected. **b**, Typical fluctuation events consist of electromagnetic components below the electron cyclotron frequency and electrostatic components in a higher-frequency range.

range. In addition to the imaging of X-ray intensity, the X-ray camera can also be used for the measurement of the two-dimensional electron temperature profile using the photon counting mode to analyze the incident X-ray photon energy. In this work, however, we operated the CCD camera with a photon counting mode and measured the count rate of incident X-ray photons to show the color contour in Fig. 1a. To enhance the sensitivity avoiding the pileup of the photon counting, we numerically binned $36 \times 36 = 1296$ pixels into a super-pixel and displayed the count rates per this super-pixel.

## Magnetic fluctuations

A magnetic probe (Bdot probe) for the fluctuation measurements consisted of a single loop placed at the edge confinement region of RT-1 and a coaxial transmission line. In general, when a Bdot probe with $N$ turns and a loop area of $S$ interlinks with a fluctuating magnetic field $B(t)$ and produces an output signal $V(t)$, the Fourier transform of the circuit equation gives

$$\frac{V(\omega)}{B(\omega)} = SN \frac{\omega^2(L/R + rC) + i\omega\left[(1 + r/R) - LC\omega^2\right]}{\left[(1 + r/R) - LC\omega^2\right]^2 + \omega^2(L/R + rC)}. \quad (13)$$

where $L$ is the self inductance of the loop, $C$ is the capacitance of the transmission line, and $R$ is the input resistance of the digitizer. Under low-frequency conditions of $\omega \ll 1/\sqrt{LC}$ and $\omega \ll R/L$, a well known expression for the Bdot probe amplitude ($V/B \propto i\omega$) is obtained. Because the frequency dependence is moderate when $R/L \ll \omega$, it is often a good approximation that $V(t)$ is proportional to $B(t)$ in higher frequencies.

In RT-1, magnetic measurements have been conducted using low-speed Bdot probes, focusing on rather low-frequency fluctuations associated with the so-called inward transport. In this study, we introduced Bdot probe coils with appropriate fewer turn number and surface area suitable for higher-frequency range for the study of whistler phenomenon. We calculated the temporal evolution of frequency power spectrum of magnetic field fluctuations using a short-time Fast Fourier Transform (SFFT) to capture their rapid frequency changes. From digitizer data acquired at a maximum sampling rate of 500 MHz, we typically applied a Hanning window to every 2048 data points to obtain the time evolution of the frequency.

A common issue with the Bdot probe measurements is the separation of inductive (magnetic, $B$) and capacitive (electrostatic, $E$) signals. The latter is caused by coupling between the loop line and the plasma. To separately measure both $B$ and $E$ signals, in this study, we used a relatively common approach to use a pair of magnetic loops with the same area oriented in opposite directions at the almost same position. Because the $B$ signal changes its sign while the $E$ component has a same polarity between the two loops, it is possible to separately measure the magnetic and electrostatic components of the fluctuation signals.

Examples of measurements are shown in Fig. 9. They indicate that the fluctuation activities consist of components with both electromagnetic and electrostatic characteristics. Figure 9a provides an example of a fluctuation event where the magnetic component is dominant. In this case, signals from two loops are observed to be nearly opposite phase, resulting in a rather intensive magnetic field signal. In Fig. 9b, for several fluctuation events with a longer time period, both the magnetic (Loop2 - Loop1) and electrostatic (Loop2 + Loop1) components of the fluctuations are plotted. Generally, chirping fluctuations observed at frequencies below 0.7 $f_{ce}$ are of electromagnetic nature, while rather broad fluctuations in the higher-frequency range exhibit electrostatic tendency.

## Data availability

The data that support the findings of this study are available from the corresponding author upon request.

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

## Acknowledgements

We thank Dr. Adam Deller for instructive suggestions on the density reconstruction method. This work was supported by JSPS KAKENHI Grant nos. 17H01177, 22H04936, and 22H00115, the NIFS Collaboration Research Program nos. NIFS19KBAR026 and NIFS22KIPR008.

## Author contributions

H.S. constructed and worked with magnetic probes. M.N. implemented interferometry and X-ray diagnostics, and analyzed hot-electron behavior in plasma. N.K. performed magnetic fluctuation measurements, developed and worked with data analysis system. Z.Y. designed the RT-1 dipole project, proposed laboratory experiments on chorus emission, and supervised the experiments and drafting the manuscript. All authors participated in the experimental planning and execution of experiments, contributed to drafting and improving the manuscript.

## Competing interests

The authors declare no competing interests.
