## [Peer Review File · Nature Communications]

Experimental study on chorus emission in an artificial magnetosphereREVIEWER COMMENTS

Reviewer #1 (Remarks to the Author):

The paper reports experimental observation of non-thermal electromagnetic emission from the ECR discharge plasma in a dipole magnetic field configuration. The manuscript is well written and contains detailed information about all aspects of the experiments. But there are questions, which should be answered.

The main conclusion of the work is the possibility to model the Earth magnetosphere and excitation of whistler waves. Authors discuss the origin of electromagnetic emission but does not explain the applicability to the magnetosphere. The magnetic field configuration is not the only thing needed to model. Whistler waves in the magnetosphere are excited and propagates in the density ducts. E.g., in the conclusion authors say “the “artificial magnetosphere” is a powerful tool to mimic the geospace environment in the laboratory”. I believe, that this statement could be made if authors discuss the applicability of the laboratory model of instability source to the space environment. Otherwise, this paper is more about basic plasma physics phenomena.

Next, the prove of whistler-mode emission is not complete. The presented data shows that plasma emission is observed at frequencies 15-100 MHz, which is refered as $0.1-0.7 f_{ce}$. But the location of the source region has never been discussed. From the manuscript it follows that the source of waves is local, since the reference electron cyclotron frequency is constant. One may conclude that $f_{ce} = 150$ MHz, which corresponds to the magnetic field strength 5 mT – the value of the magnetic field strength near the chamber wall, where b-dot probes are located. Whistler waves propagates only in dense plasma, where $f_{pe} > f_{ce}$. From the text (only one mention at page 9) it follows, that minimal $f_{pe} = 1$ GHz near vacuum chamber walls. So, whistler waves could propagate. But the source location could be different. Moreover, the waves are excited by the energetic electron fraction, which is localized closer to the magnetic coil. To confirm the mode type of detected plasma emission please provide the information of the wave source location, plasma density profile and the ratio of f_{pe}/f_{ce} over it. Also, discuss on the type of the source – localized or distributed one.

The above-mentioned value of $f_{ce}=150$ MHz is the smallest one in the system and could be only bigger. Why the spectrum of electromagnetic emission is cut at 150 MHz? Are there any other emissions at higher frequencies? What is the Nyquist frequency?

Another evidence of the whistler mode excitation is the analysis of the wave dispersion relation. But there are only 7 data points in the Fig. 4d, which is not enough to confirm that. Also, it is not clear what is the average parameters (average over what?) of the setup for the line (1) and why authors use it.

Please, provide more statistics here. As I can see from the paper, the measured wave vector is k_z . If so, please comment on the relation between experimental data and wave dispersion relation.

Since the waves are excited by the energetic electron fraction, please, provide more information on the electron energy distribution function. Having this information, please compute the estimate of the whistler wave instability increment and prove that it has reasonable value for the experimental parameters.

Other comments:

- 1) page 8. "At least some of these higher frequency activities show a positive correlation with the occurrence of strong electromagnetic fluctuations in the 0.1 - 0.4 f_{ce} range." How this conclusion was made and what is the consequence?
- 2) page 14, last paragraph. Typo – "growth rate α of".
- 3) page 16, line after formula (2). Typo in the definition of f_{ce} .
- 4) page 17. Please add a reference what is the Grad-Shafranov numerical analysis?
- 5) Add the description how the spectrum of wave activity was computed at the end of the section Diagnostics.
- 6) ExtendedFig1. Loop1+Loop2 plot is almost zero. Please change the vertical axis ratio to fix it, or provide another shot.
- 7) ExtendedFig2. The information showed in this figure is very important to understand the method of separation of two different wave types. But is not described in the text. Please add more information to the text of the manuscript.
- 8) Every experimental figure with the wave field oscillations and its spectrum should be accompanied by the information about the relevant plasma parameters in figure caption, such as, n_e , f_{pe} , f_{ce} .

Reviewer #2 (Remarks to the Author):

The manuscript from Saitoh et al. on the reproduction of chorus emission in an artificial magnetosphere is innovative and worth being published by Nature communications. It is the first laboratory observation of chirping whistler waves in a magnetic geometry, similar to the geometry of magnetospheres. The comparison of typical plasma parameters in the RT-1 experiment and the geospace environment is particularly enlightening. However, I have a number of remarks that I would like to see taken into account

Major

1. p. 4, The Soft X-ray image in figure 1 has no scale hence impossible to interpret other than it's blue, green or red.
2. p.7, The experiment lasts 3 seconds but in figure 3, a zoom on 3 ms is presented without mentioning when/where during the 3 seconds, this should be highlighted in figure 2
3. p.7, In figure 3b 4 zooms extracted from Figure 3a are presented with the following time periods
 1. 2.28-2.33 ms, so within the 3 ms presented
 2. 41.19-41.23 ms, so outside the 3 ms presented
 3. 32.5-32.65 ms, so outside the 3 ms presented
 4. 82.43-82.46, so outside the 3 ms presented
5. So please provide an explanation and highlight the 4 time periods to clearly link figure 3a and 3b
4. p.7, the manuscript states : figure 3 show the detailed temporal evolution of each of fluctuation events that I understood as $t=0-0.1s$ and $t= 2-3s$ but figure 3 displays one time period of 3 ms, hence only one event, so which one?
5. One important point that is not clear in this version of the manuscript is what in the setup and/or in the diagnostics of RT-1 makes these chorus like waves now visible?

Inappropriate general statements

1. p.1, The abstract states that geospace activities critically affect advanced technologies as well as climate change is to me inappropriate as climate change as nothing to do with this study so why mentioning it?
2. p.2, the importance of space weather phenomena has necessitated active research in this area. This sentence is not necessary as chorus has been actively studied both observationally and theoretically since their discovery and particularly for the past 20 years, so it is not space weather which has necessitated active research in this area.

Minor

- p.3, RT-1 (Ring Trap-1) shall be defined the first time it is mentioned on p.3 and not on p.4
- p.5, reference to Figure 2b shall be Figure 2c
- P.5, reference to Figure 2c shall be Figure 2d

Minor typos

p.2 please correct to proper English the following sentence: helping the understand

p.4, caption of Figure 1, first word shall appear as "Inside" not "Inis de"

p.5, please correct "on the order"

p.10, please replace "geospacer" by "geospace"

Reviewer #3 (Remarks to the Author):

May 2, 2023

Review of

Reproduction of chorus emission in an artificial magnetosphere by

H. Saitoh, M. Nishiura, N. Kenmochi, and Z. Yoshida

This paper summarizes experiments on the levitated dipole experiment RT-1 where spontaneous generation of chirping whistler waves are observed. The paper identifies the free energy source as temperature anisotropy produced through electron cyclotron resonance heating. The paper reports on the observations of the frequency and wave-vector measurements and their toroidal coherence in the device. The paper is well written and very exciting scientific progress. I recommend publication. I would encourage the authors to make two minor additions to the paper to help put this experiment in the proper context.

First, laboratory studies of chorus like phenomenon were also performed at the US Naval Research Laboratory in a linear device (*e.g.* [1]) where both spontaneous emission and stimulated emission (or triggered) chorus was studied.

Second, in figure 4 the measured wavelength was reported as between 1 and 0.3 meters which is comparable to the size of the plasma (also reported in Figure 4). This has strong implications for the generation mechanism here since the wavelength is comparable to the system gradient scale size. This is in contrast to previous laboratory experiments (at UCLA and NRL) and in contrast to the Earth's radiation belts where WKB approximations are more appropriate.

References

- [1] E. M. Tejero, C. Crabtree, D. D. Blackwell, W. E. Amatucci, G. Ganguli, and L. Rudakov. Experimental characterization of nonlinear processes of whistler branch waves. *Physics of Plasmas*, 23(5):055707, 2016.

Replies to Reviewer #1:

We would like to thank the reviewer for the careful and insightful comments that have helped our substantial progress to improve the manuscript. Studying the critical comments of the referee, we admit that the original manuscript (and the title) do not explain the purpose of the present work properly, and some bold statements obscured the conclusion. As pointed out, our artificial magnetosphere does not completely reproduce the Earth's magnetosphere in many aspects. However, it contains the minimum set of elements that can produce the chorus emission. As one of the aims of physics is to derive "minimum models" of phenomena in nature, our purpose is to study the chorus emission in an abstract magnetosphere that possesses essential elements characterizing a larger set of similar systems, ranging from planetary magnetospheres to high-energy astrophysical objects, as well as fusion plasma systems.

In this reply letter, we write the reviewer comments in red, our replies in blue, and text in the manuscript in green.

1. The paper reports experimental observation of non-thermal electromagnetic emission from the ECR discharge plasma in a dipole magnetic field configuration. The manuscript is well written and contains detailed information about all aspects of the experiments. But there are questions, which should be answered.

We are delighted to read this concise summary of the novelty of our manuscript. For addressing the raised concerns, we have conducted a series of additional experiments. Furthermore, we have restructured the manuscript, avoiding confusion while emphasizing the aspects of our research that we believe are crucial. Revisions made and point-by-point replies to the comments are summarized as follows.

2. The main conclusion of the work is the possibility to model the Earth magnetosphere and excitation of whistler waves. Authors discuss the origin of electromagnetic emission but does not explain the applicability to the magnetosphere. The magnetic field configuration is not the only thing needed to model. Whistler waves in the magnetosphere are excited and propagates in the density ducts. E.g., in the conclusion authors say “the “artificial magnetosphere” is a powerful tool to mimic the geospace environment in the laboratory”. I believe, that this statement could be made if authors discuss the applicability of the laboratory model of instability source to the space environment. Otherwise, this paper is more about basic plasma physics phenomena.

Thank you for this comment. As pointed out, the Earth's magnetosphere is a vast and complex system, and modeling the entire processes within the magnetosphere in a laboratory remains an unattained goal. We agree with the reviewer on this point. Before addressing the improvement made in the manuscript according to the comments, we would like to explain the general purpose of the RT-1 project and the current research perspective on this context.

We wish to focus on the fact that magnetic dipoles (or those globally approximated as dipoles) exist universally in space and laboratory environments. These examples include planetary magnetospheres, high-energy astrophysical objects, and fusion plasma experiments, where common fundamental physics should exist. In particular, the interaction of particles with waves, such as chorus emissions, commonly plays important roles in a broad range of physical phenomena. In dipole fusion plasmas, for example, transport resulting from the interaction between charged particles and waves is a key factor in achieving excellent confinement and realization of fusion power production. Interactions between waves and particles can excite different waves, whose nonlinear interactions drive various phenomena including turbulence, self-organization, and chaotic behaviors. We consider these diverse issues as "physics of magnetospheric plasmas" in a broader sense, which are relevant to a wide range of nonlinear plasma phenomena. In this context, the ultimate goal of RT-1 is to elucidate general physical phenomena of plasmas in a highly non-uniform magnetospheric configuration. On the present work, this manuscript is an experimental study on the excitation of chorus emissions, which revealed that these wave activities appear ubiquitously in laboratory and space environments when plasma satisfies certain conditions. We believe that such achievement in a controlled environment using a magnetically levitated superconducting coil is a substantial progress in its own right, and represents an important step forward not only in fundamental plasma physics but also in space plasma science.

In page 1,

we modified the title from "**Reproduction of chorus emission in an artificial magnetosphere**" to "**Experimental study on chorus emission in an artificial magnetosphere**" so that our aforementioned research motivation and present work are clearly described.

In the abstract, we deleted "**which mimics the geospace environment in a laboratory**", which was an additional explanation for the levitated dipole experiment, to avoid confusion.

Also in the abstract, we deleted "**Because geospace activities critically affect the advanced**

technologies as well as climate change, there is an urgent need to establish reliable space weather modelling based on an accurate understanding of wave particle interactions in space." to avoid overstatement concerning the present scope of the study.

In page 2, "A levitated superconducting ring coil can create an artificial magnetosphere that mimics planetary magnetospheres" was modified to "A levitated superconducting ring coil creates a dipole magnetic field that is common to planetary magnetospheres."

In the conclusion, in pages 12 and 13, we deleted "'the artificial magnetosphere" is a powerful tool to mimic the geospace environment in the laboratory" to avoid overstatement, and modified the concluding and perspective remarks as follows:

In this work, spontaneously excited chirping electromagnetic and rather broadband electrostatic waves were experimentally investigated in the RT-1 levitated dipole. Chorus emission has been shown to appear ubiquitously in laboratory and space environments for various parameters when plasma satisfies certain conditions. Investigations conducted in a controlled laboratory environment with higher reproducibility and high-resolution diagnostics are expected to provide a new complementary approach to further understanding the chorus emission phenomena that are common in both laboratory and geospace plasmas. In the future, this research might contribute to the understanding of wave particle interactions in the space weather system, which has a critical impact on advanced technologies and climate change [1].

3. Next, the prove of whistler-mode emission is not complete. The presented data shows that plasma emission is observed at frequencies 15-100 MHz, which is refered as $0.1-0.7 f_{ce}$. But the location of the source region has never been discussed. From the manuscript it follows that the source of waves is local, since the reference electron cyclotron frequency is constant. One may conclude that $f_{ce} = 150$ MHz, which corresponds to the magnetic field strength 5 mT – the value of the magnetic field strength near the chamber wall, where b-dot probes are located. Whistler waves propagates only in dense plasma, where $f_{pe} > f_{ce}$. From the text (only one mention at page 9) it follows, that minimal $f_{pe} = 1$ GHz near vacuum chamber walls. So, whistler waves could propagate. But the source location could be different. Moreover, the waves are excited by the energetic electron fraction, which is localized closer to the magnetic coil. To confirm the mode type of detected plasma emission please provide the information of the wave source location, plasma density profile and the ratio of f_{pe}/f_{ce} over it. Also, discuss on the type of the source – localized or distributed one.

We appreciate the mention of these points, as the propagation and source of the emission are

crucial issues. Here we answer to the issues on f_{ce} and f_{pe} based on the additional experiments and analysis. Concerning the dispersion relation and type of the source, we would like to answer these issues in our reply to the question Nos. 4 and 5.

With respect to the conditions of wave propagation and the source region, we conducted experiments to obtain the relationship between f_{ce} and f_{pe} by evaluating the density distribution in the confinement region. As suggested by the reviewer, electron density in the vicinity of the plasma edge, where the chorus emission fluctuations were detected, is a crucial parameter in this study. Therefore, in addition to microwave interferometers that have been provided the information of line-integrated electron density, we installed edge Langmuir probes, which are capable of operating in high-beta plasmas. Using a technique of a global optimization method, we reconstructed the electron density distribution and evaluated the f_{pe}/f_{ce} profile. As shown in the newly added Fig.3, the results indicate that, except for the very vicinity of the levitating coil, f_{pe}/f_{ce} exceeds one over a wide region within the plasma, indicating the chorus waves to propagate in most of the plasma interior.

Fig.3 Several plasma parameters in the RZ cross section of RT-1. a, electron density n_e , b, plasma frequency f_{pe} c, electron cyclotron frequency f_{ce} , and d, their ratio f_{pe}/f_{ce} . Pale blue thin lines in d shows the magnetic surfaces generated by the levitated superconducting coil and lifting coil.

In page 7, we added the following explanations in the text together with the new Fig.3.

Through the following experiments, unless otherwise mentioned, we generated plasma in RT-1 under the conditions of 11 kW of 2.45 GHz ECH power injection and an initial filled helium gas pressure of 2.5 mPa. For these experimental conditions, electron density profile was obtained using a global optimization method from measurements with microwave interferometers and an edge Langmuir probe. Figure 3 shows the spatial profiles in the RZ cross section of RT-1, including the electron density n_e , electron plasma frequency f_{pe} , electron cyclotron frequency f_{ce} , and their ratio f_{pe}/f_{ce} . Except for the strong magnetic field and low electron density region in the vicinity of the levitated superconducting coil, f_{pe}/f_{ce} exceeds 1 over a wide region within the plasma, indicating the chorus waves to propagate in most of the plasma interior.

In page 3, we also revised Table 1, on the parameters of RT-1, based on the above updated results.

Also in the Methods section, from page 19, we added the following short explanations on the method to estimate n_e and f_{pe}/f_{ce} .

In order to assess information regarding the transmission conditions and sources of whistler wave propagation throughout the entire plasma, we performed a density distribution reconstruction using a global optimization technique. To incorporate information from the plasma edge region where fluctuation measurements were taken, we included new information from the edge Langmuir probes in the evaluation function. We then used the difference between the computed density distribution and the experimental data, represented with multiple parameters, to search for a density profile solution with the minimum of an evaluation function value in the entire computational domain.

Once the search for a single solution converged to a local minimum, we utilized an algorithm that searches for local minima again from initial values that are randomly displaced from the previous one in both direction and distance. This approach allows us to explore local minima across the entire computational region. EctendedDataFig.1 illustrates the convergence of the global optimization algorithm for a test evaluation function F set for a case with two variables x_1 and x_2 , successfully detecting all the local minimum values. From the detected local minimum values, we selected the most suitable one as the solution.

MethodsFig.1: Example of search of solution in the technique of a global optimization method. Starting from randomly distributed initial values (blue crosses), all local minimum values (red circles) are detected in the entire computational region.

In order to accurately reflect information from the plasma edge where oscillations were measured, we conducted a search for solutions using measured interferometer data, I_{mi} , along with electron density measurements, n_m , obtained using edge probes near the vacuum vessel walls. Initially, on the equatorial plane of the device, we provided density in the region $\psi_1 \leq \psi \leq \psi_2$ between the coil surface (magnetic flux function $\psi = \psi_1$) and the vacuum vessel wall ($\psi = \psi_2$) as:

$$n(r, z = 0) = n_0 |\psi - \psi_1| |\psi - \psi_2|^a$$

As illustrated in MethodsFig.2, by choosing the values of parameters n_0 and a , it is possible to select the absolute value of density and the shape of the density distribution within the confinement region with high flexibility.

MethodsFig.2: Shape of electron density profile on the equator of the device. By changing a , various shapes of density profiles are represented.

On the same magnetic field line, we accounted for the dependence on the magnetic field by calculating:

$$n(r, z) = n(r, z = 0) \times \left(\frac{B}{B_0}\right)^b$$

where $B_0 = B(r, z = 0)$ is the field strength on the equator and $B(r, z)$ is that of each of the measurement point. This method allowed us to incorporate the dependence on the magnetic field into the density distribution on the RZ plane. We then computed quantities that can be obtained through measurements. These include the line-integrated density in the line of sight as measured by the microwave interferometer:

$$I_{ci} = \int n(r, z) dl_i$$

and the local density measured at the same radial position as the Bdot probe with the Edge Langmuir probe:

$$n_{ce} = n(r = 0.98, z = 0)$$

With these quantities, we reconstructed the density by searching for parameters that minimize the function:

$$F = \sum_i (I_{ci} - I_{mi})^2 + (n_{ci} - n_{mi})^2$$

We conducted the search for parameters with experimental measurements of interferometer results: $I_{m1} = 2.234 \times 10^{17} \text{ m}^{-2}$, $I_{m2} = 7.056 \times 10^{16} \text{ m}^{-2}$, $I_{m1} = 3.203 \times 10^{16} \text{ m}^{-2}$, and edge density $n_{r=0.98} = 3.012 \times 10^{15} \text{ m}^{-3}$. The search for solutions resulted in the best-fit parameters: $n_0 = 2.347 \times 10^{11}$, $a = 1.082$, and $b = 0.6375$. The electron density n_e and the corresponding electron plasma frequency f_{pe} distributions in the RZ section of the RT-1 device are shown, along with the electron cyclotron frequency f_{ce} with respect to the magnetic field configuration (f_{pe}/f_{ce}).

We also appreciate the comments regarding X-ray images of high-energy electrons and hot electron distribution. As shown in the revised Fig. 1 (the applicable part is shown below, left), the field of view of the X-ray camera is limited to the vicinity of the coil. Therefore, this image does not mean that the distribution of high-energy electrons is localized solely near the magnetic coil. In the present experimental setup, unfortunately, it is not applicable to further extend the field of view of the X-ray camera. Both equilibrium analysis of pressure distribution and past X-ray measurements with Si(Li) and CdTe detectors indicate that the high-energy electrons responsible for plasma pressure are distributed over a broad range within the plasma confinement region. Also, past experiments have further indicated the presence of high-energy

electrons in the peripheral regions, as evidenced by the melting of a stainless steel (SUS304) wall probe introduced at the edge plasma region, due to heat flux from high-energy electrons (figure below, right). Therefore, it would not be unnatural that hot electrons do exist near the vacuum chamber wall and they can cause the observed emission.

(left) On X-ray camera field of view from Fig.1. (right) Edge wall probe melted because of high energy hot electrons located in the peripheral region close to the chamber wall.

To avoid unclearness and confusion on this point associated with the emission source, we have made additional explanations and modifications to the figures. In Fig.1 (1) in page 4, a circle to show the field of view and additional explanation in the caption "Soft X-ray image in the relatively strong field region" were added. In the main text, an explanation on the field of view of the camera was added as follow:

(in page 5)

Measurements with Si(Li) and CdTe detectors confirmed the presence of high-energy electrons over a wide region within the plasma, including the edge region near the vacuum chamber wall.

4. The above-mentioned value of $f_{ce}=150$ MHz is the smallest one in the system and could be only bigger. Why the spectrum of electromagnetic emission is cut at 150 MHz? Are there any other emissions at higher frequencies? What is the Nyquist frequency?

We appreciate this comment. We did not observe clear fluctuation activities above approximately 150 MHz. The sampling frequency of the measurement used was up to 500 MHz. In MethodsFig.4, the frequency spectrum range has been extended to the Nyquist frequency of 250 MHz for display. Fluctuations are primarily detected in the region below

around 100 MHz, but are rarely observed beyond 150 MHz. As indicated in Fig.3, the electron cyclotron frequency in the strong magnetic field region exceeds 10 GHz. If the whistler emission occurred in such a stronger magnetic field region and transmitted from there, it is highly unnatural that there is an almost complete absence of detectable fluctuation activities above ~ 100 MHz (1% of the electron cyclotron frequency of the strong field region). Based on these observations, it would be reasonable to interrupt that the source of fluctuations is localized in the weak magnetic field region at the edge plasma.

We added these measurement results together with additional explanations, in page 24 as well as in pages 10 and 11:

As indicated in Fig.4 and MethodsFig.4, fluctuations are primarily detected in the region below around 100 MHz, but are rarely observed beyond 150 MHz. As shown in in Fig.3, on the other hand, the electron cyclotron frequency exceeds 10 GHz in the stronger field region of RT-1. If the whistler emission occurred in such a stronger magnetic field region, transmitted, and detected at Bdot probes at the plasma edge, it is highly unnatural that there is an almost complete absence of detectable fluctuation activities above ~ 100 MHz (1% of the electron cyclotron frequency of the strong field region). Based on these observations, we anticipate that the source of fluctuations is localized in the weak magnetic field region at the edge plasma.

MethodsFig.4: Temporal evolution of power spectrum of fluctuation activity. Time series of voltage signal measured using a Bdot probe with a sampling rate of 500MHz was transformed to spectrogram by STFFT. For detailed structures of the fluctuation, refer Fig.4 in the main text. Constant frequency peaks appeared at 19, 25, 58, 192, and 225 MHz are caused by environmental noise.

5. Another evidence of the whistler mode excitation is the analysis of the wave dispersion relation. But there are only 7 data points in the Fig. 4d, which is not enough to confirm that. Also, it is not clear what is the average parameters (average over what?) of the setup for the line (1) and why authors use it. Please, provide more statistics here. As I can see from the paper, the measured wave vector is k_z . If so, please comment on the relation between experimental data and wave dispersion relation.

Thank you for this comment. Concerning the dispersion relation and the propagation behavior, we added a new measurement result and explanations based on diagnostics with newly introduced Bdot probes, suitable for phase difference detection, and an edge Langmuir probe. The previous magnetic fluctuation measurement system did not primarily aim to obtain dispersion relations, and a remaining challenge was the difficulty in determining the phase difference of fluctuations due to the relatively wide spacing between measurement points along the magnetic field lines. That is the reason why we could provide only 7 data points with confident. Furthermore, for calculating the dispersion relations, electron density values obtained through microwave interferometers' line integrations and estimations were used in the previous figure. We agree with the reviewer's remarks that these were not suitable for accurately determining the dispersion relations. To address these issues, we conducted phase difference measurements using new Bdot probes with smaller spacing between measurement points along the magnetic field lines to accurately measure phase differences, and compare them with Whistler dispersion relations. Here we directly measured electron density using a new Langmuir probe, mentioned in the previous answer, at the same radial position as the Bdot probe, incorporating this data into the dispersion relation calculations. The figure illustrates the comparison between the obtained dispersion relations and the calculated values obtained in this manner. We replaced the previous figure for dispersion relation in Fig.5d with this new figure and added the following explanations, in page 10.

(in page 11)

With measurements on the same poloidal cross-section in Fig.5c, we detected the propagation of fluctuation wave along magnetic field lines. ... The parallel wave numbers inferred from measurements with separately placed Bdot probe with a vertical distance of 1 cm along field line are plotted for waves with different frequencies, as shown in Fig.5d. The dispersion relations of the whistler mode calculated for the magnetic field strength and electron density at the fluctuation measurement position of plasma edge, $B = 5.4$ mT and $n_e = 6.3 \times 10^{14} \text{ m}^{-3}$ is shown as a solid line in Fig.4d, which shows relatively good agreement with the measurement results.

Fig.5. d, Dispersion relation for local electron density and magnetic field strength at the Bdot probe position (line) and measured values of wave number and frequency (circles).

It is also noted that we conducted experiments with different filling gas (hydrogen and helium) and saw no clear change in the appearance frequency range of these fluctuation activities, supporting that they are caused by electrons in the plasma.

6. Since the waves are excited by the energetic electron fraction, please, provide more information on the electron energy distribution function. Having this information, please compute the estimate of the whistler wave instability increment and prove that it has reasonable value for the experimental parameters.

In addition to bulk (low temperature) electron measurements that can be obtained using edge Langmuir probes, we measured the temperature of high-energy electron components responsible for instabilities using X-ray wave-number analysis relative to the current parameters and incorporated information about the energy distribution into the paper. As depicted in the figure, the high-temperature electron temperature within the plasma for the experimental parameters in this study was approximately 9 keV. We added the following description in the Method section, in page 23.

In RT-1, the hot electron component have been measured with the Pulse height analysis (PHA) method using Si(Li) and CdTe detectors. In the operation conditions presented in this work, ECH microwave power of 11 kW and filling helium gas pressure of 2.5 mPa, we measured the

temperature of the high-energy electron component, primarily responsible for plasma pressure, using a Si(Li) detector. When electrons follow a Maxwellian distribution with a temperature, T_e , the continuous spectrum emitted from the plasma can be described in terms of the number of photons per unit energy as

$$\frac{dN}{dE_x} = \alpha n_e n_i \frac{g_f \exp(-E_x/T_e)}{E_x \sqrt{T_e}},$$

where g_f is the Gaunt factor, n_e is electron density, n_i is ion density, and α is a constant. The figure represents the energy spectrum of X-rays obtained in this manner, and the high-energy electrons responsible for plasma pressure are estimated to have a temperature of approximately 9 keV. The relatively good linearity of X-ray intensity with photon energy suggests that these high-energy electrons have a distribution rather close to a Maxwellian distribution.

MethodsFig.3: X-ray energy spectrum emitted from the hot electrons of RT-1 plasma. Electrons consist of bulk cold component and energetic components.

On the effects of these hot electrons in the plasma, we expanded the discussion on instabilities related to these high-temperature electrons and connected it to the measurement of hot electrons, in page 8:

The most active mode appears in the frequency range of $f = 15 - 55$ MHz (approximately $0.1 - 0.4 f_{ce}$). As explained in the Methods section with MethodsFig.3, plasma has a hot electron component with approximately 9 keV in the present formation conditions. In the high- β plasma generated by ECR heating, electrons have temperature anisotropy, especially near the equator of the dipole field. When we assume that the upper limit of this most active frequency range is set by the condition for an unstable whistler frequency in the linear theory [9,10] of

$$f < f_{ce} A / (A + 1), \quad (1)$$

the electron temperature anisotropy is estimated to be $A = T_{\perp}/T_{\parallel} - 1 = 0.7$, which is fairly consistent with Grad-Shafranov equilibrium analysis including the temperature anisotropy [36-38].

7. Other comments:

We deeply thank the reviewer for careful reading and valuable suggestions. We revised the manuscript accordingly.

1) page 8. “At least some of these higher frequency activities show a positive correlation with the occurrence of strong electromagnetic fluctuations in the 0.1 - 0.4 f_{ce} range.” How this conclusion was made and what is the consequence?

By further examining the experimental data on this point, we came to a conclusion that there is a need for additional experiments and further consideration regarding the intervals between the appearance of different types of fluctuations and the duration of quiescent periods. To avoid including rather vague descriptions on this aspect in this work, we deleted this sentence.

2) page 14, last paragraph. Typo – “growth rate α of”.

We added spaces before and after α in page 17.

3) page 16, line after formula (2). Typo in the definition of f_{ce} .

We corrected the part where " f_{pe} " was erroneously written as " f_{ce} ." in page 19.

4) page 17. Please add a reference what is the Grad-Shafranov numerical analysis?

We added a brief explanation on the Grad-Shafranov analysis in the Methods section along with additional references including one with a relatively good availability, in page 22.

The Grad-Shafranov equation provides magnetohydrodynamic equilibria for axisymmetric plasmas, and solving it numerically allows us to obtain the equilibrium distribution of plasma [36-38].

[36] Furukawa, M. Effects of pressure anisotropy on magnetospheric magnetohydrodynamics equilibrium of an internal ring current system. *Phys. Plasmas* 21, 012511 (2014).

[37] Grad, H., & Rubin, H. Hydromagnetic equilibria and force-free fields. *Proc. 2nd UN International Conference on Peaceful Uses of Atomic Energy*, 31, 190 (1958).

[38] Freidberg, J. P., *Ideal MHD* (Plenum Pub Corp, 1987).

5) Add the description how the spectrum of wave activity was computed at the end of the section

Diagnostics.

We added the following explanation on the power spectrum measurement using short-time FFT in the Diagnostics section, in page 25.

As shown in MethodsFig.4, we calculated the temporal evolution of frequency power spectrum of magnetic field fluctuations using a short-time Fast Fourier Transform (SFFT) to capture their rapid frequency changes. From digitizer data acquired at a maximum sampling rate of 500 MHz, we typically applied a Hanning window to every 2048 data points to obtain the time evolution of the frequency.

6) ExtendedFig1. Loop1+Loop2 plot is almost zero. Please change the vertical axis ratio to fix it, or provide another shot.

(left) Previous MethodsFig.5 (previous ExtendedFig1), and (right) revised figure.

We modified this figure accordingly, from the (left) to the (right) of the following figures. Previously, we have intended to show that the dominant component of this fluctuation in this figure was the magnetic component. We judged that this should be explained with the following figure (MethodsFig.5), rather than this figure (MethodsFig.6) and revised the figure accordingly, in page 25.

7) MethodsFig5. The information showed in this figure is very important to understand the method of separation of two different wave types. But is not described in the text. Please add more information to the text of the manuscript.

Thank you for this instructive comment. In the explanation of Bdot probe, after the explanation of the measurement principles, we have added information that can be inferred from the results presented in two figures, in page 26.

Examples of measurements are shown in MethodsFigs. 5 and 6. They indicate that the fluctuation activities consist of components with both electromagnetic and electrostatic characteristics. MethodsFigs. 5 provides an example of a fluctuation event where the magnetic component is dominant. In this case, signals from two loops are observed to be nearly opposite phase, resulting in a rather intensive magnetic field signal. In MethodsFig. 6, for several fluctuation events with a longer time period, both the magnetic (Loop2 - Loop1) and electrostatic (Loop2 + Loop1) components of the fluctuations are plotted. Generally, chirping fluctuations observed at frequencies below $0.7 f_{ce}$ are of electromagnetic nature, while rather broad fluctuations in the higher-frequency range exhibit electrostatic tendency.

8) Every experimental figure with the wave field oscillations and its spectrum should be accompanied by the information about the relevant plasma parameters in figure caption, such as, n_e , f_{pe} , f_{ce} .

We thank the reviewer for the suggestion to improve the manuscript on this point. As already explained in the answer to question No. 3, we have added the following explanation on plasma parameters and generation conditions as common parameters for several experiments.

(This is just a repeat of the previous text explained at the reply to question No. 3., In page 7)
Through the following experiments, we generated plasma in RT-1 under the conditions of 11 kW of 2.45 GHz ECH power injection and an initial filled helium gas pressure of 2.5 mPa. For these experimental conditions, electron density profile was obtained using a global optimization method from measurements with microwave interferometers and an edge Langmuir probe. Figure 3 shows the spatial profiles in the RZ cross section of RT-1, including the electron density n_e , electron plasma frequency f_{pe} , electron cyclotron frequency f_{ce} , and their ratio f_{pe}/f_{ce} . Except for the strong magnetic field and low electron density region in the vicinity of the levitated superconducting coil, f_{pe}/f_{ce} exceeds 1 over a wide region within the plasma, indicating the chorus waves to propagate in most of the plasma interior.

Due to the strong non-uniformity of the magnetic field, it is not straightforward to write the typical values of experimental parameters. So we decided to refer the spatial profiles of these values. Because experiments in Figs. 4 and 5 were conducted with common conditions to that

of Fig.3, we added the following description of experimental conditions in the figure captions therein, so that a reader can refer the above explanation and the information of n_e , f_{pe} , and f_{ce} shown in Fig.3 for reading the plasma parameters, in page 7.

Plasma was generated with $P_f = 11$ kW of 2.45 GHz ECH power and an initial filled helium gas pressure of $P_n = 2.5$ mPa.

In Fig.6 in page 12, we also added the following plasma formation conditions in the caption.

The 2.45 GHz ECH power P_f was between 1 and 19 kW, and the initial filled helium gas pressure P_n was between 1.5 and 7.5 mPa.

Thank you very much for the detailed and insightful review on the manuscript.

Replies to Reviewer #2:

We would like to thank the reviewer for recognizing the value of the present work. We sincerely appreciate the insightful review comments that gave us constructive help to substantially improve the manuscript. We studied your comments carefully and revised the manuscript accordingly. We write the reviewer comments in red, our replies in blue, and text in the manuscript in green.

1. The manuscript from Saitoh et al. on the reproduction of chorus emission in an artificial magnetosphere is innovative and worth being published by Nature communications. It is the first laboratory observation of chirping whistler waves in a magnetic geometry, similar to the geometry of magnetospheres. The comparison of typical plasma parameters in the RT-1 experiment and the geospace environment is particularly enlightening. However, I have a number of remarks that I would like to see taken into account

Thank you very much for your kind words. We would like to send point-by-point replies in the following text.

Major

1. p. 4, The Soft X-ray image in figure 1 has no scale hence impossible to interpret other than it's blue, green or red.

We appreciate this comment to improve the visibility of the figure. The color contour plot represents the count rate of X-ray photons observed at each of the pixels shown. We have revised the figure to include a scale, and added an explanation in the text clarifying that the measurement results are in terms of count rates.

We also added a short explanation on the usage of the CCD camera for this measurement in the figure caption of Fig.1, in page 4:

Soft X-ray image in the relatively strong field region, shown as a colour contour map of the flux count of incident photons,

as well as additional descriptions below in the Methods section, in pages 23 and 24:

In this work, however, we operated the CCD camera with a photon counting mode and

measured the count rate of incident X-ray photons to show the colour contour in Fig.1a. To enhance the sensitivity avoiding the pileup of the photon counting, we numerically binned $36 \times 36 = 1296$ pixels into a super-pixel and displayed the count rates per this super-pixel.

On X-ray camera field of view from Fig.1, in page 4.

2. p.7, The experiment lasts 3 seconds but in figure 3, a zoom on 3 ms is presented without mentioning when/where during the 3 seconds, this should be highlighted in figure 2

We appreciate this comment of the reviewer to avoid confusion in the manuscript. As pointed out, in Fig.2, we showed the results of a rather long-time (3 s) discharge period to clearly demonstrate the correlation between the diamagnetic signal (associated with the plasma pressure due to hot electrons) and the fluctuation activities. However, to study the detailed characteristics of fluctuations in Fig.4 (previous Fig.3), including the temporal evolution of frequency, sampling rate of the data typically above 300 MS/s is needed. It is not feasible for the present our measurement system to record the entire discharge period (longer than 1 s) with such a high sampling rate. Meanwhile, it would not absolutely necessary to do so for the study of each of the fluctuation events. Because of these reasons, we conducted measurements with different sampling rates for different plasma discharge shots for Fig.2 and Fig. 4. On this basis, we revised the manuscript so that the experimental conditions become clearer. While we cannot highlight each of the events because of the aforementioned reason, we revised the manuscript as follows, to avoid confusion between Figs. 2 and 4.

In page 8, we started the explanation of Fig.4 as

"Figure 4 shows the detailed temporal evolution of each of fluctuation events that were

measured with higher sampling rate than that of Fig 2."

and modified the following sentence from

"The wave activity shown in Fig.2e consists of ..." to

"The wave activity consists of ...".

in page 8, just after the above correction.

Also in the figure caption of Fig.4, in page 9, we modified

"The fluctuations in Fig.2e consist of abrupt and repeated discrete fluctuation events." to

"The fluctuation activities consist of abrupt and repeated discrete fluctuation events."

We believe that the experimental conditions are now clearly explained in the manuscript.

3. p.7, In figure 3b 4 zooms extracted from Figure 3a are presented with the following time periods

1. 2.28-2.33 ms, so within the 3 ms presented

2. 41.19-41.23 ms, so outside the 3 ms presented

3. 32.5-32.65 ms, so outside the 3 ms presented

4. 82.43-82.46, so outside the 3 ms presented

5. So please provide an explanation and highlight the 4 time periods to clearly link figure 3a and 3b

4. p.7, the manuscript states : figure 3 show the detailed temporal evolution of each of fluctuation events that I understood as $t=0-0.1s$ and $t= 2-3s$ but figure 3 displays one time period of 3 ms, hence only one event, so which one?

Thank you for these detailed and careful comments on Fig.4 (previous Fig.3) to avoid problems caused by the similar reasons to the previous point. We agree with the reviewer that this point was again not clear in the manuscript and appreciate your comments on this matter. Because the types of fluctuation events are diverse, as shown in the figures, it was impossible to find all these typical events from a single shot shown in Fig.3a. Therefore, measurements in Fig.4b were conducted for different plasma discharge shots with the same plasma formation conditions. To improve the manuscript on this point, we revised the manuscript and explained that the purpose of Fig. 4b is to illustrate the temporal evolutions of typical fluctuation events, rather than illustrating the enlarged views from Fig.4a, and conducted repeated measurements. We added the following explanations in the figure caption of Fig.4.

(in page 9, at the beginning of the caption of Fig.4b)

The temporal evolutions of typical fluctuation events measured with same plasma formation

conditions but different plasma discharge shots. These events are classified into down chirping, up chirping, multiple repeat of down and up chirping, and simultaneous appearance of different wave events.

We thank again for the careful comments on this point, and we believe that the measurement conditions of the waveforms are now clearly presented in the revised manuscript.

5. One important point that is not clear in this version of the manuscript is what in the setup and/or in the diagnostics of RT-1 makes these chorus like waves now visible?

We are grateful for this advice on making the manuscript more understandable. The detection of chorus-like emissions was made possible by the introduction of a high-speed Bdot probe. In RT-1, magnetic measurements have traditionally been conducted using diamagnetic loops and low-speed Bdot probes, focusing on rather low-frequency fluctuations associated with the so-called inward transport, typically at around 1 kHz.

The Whistler waves measured in this study exhibit frequencies that are 10^3 to 10^5 times higher than these low-frequency oscillations. Because the output voltage of the Bdot probe is proportional to the fluctuation frequency, coils with many turns are often used to identify these weak low-frequency signals. We could not detect higher frequency fluctuation activities with such Bdot probes due to the increased inductance of the measurement circuit. By introducing coils with the appropriate number of turns and surface area suitable, we are now able to observe the whistler phenomenon in this study. We added an explanation on this point as follows.

(in page 25)

In RT-1, magnetic measurements have been conducted using low-speed Bdot probes, focusing on rather low-frequency fluctuations associated with the so-called inward transport. In this study, we introduced Bdot probe coils with appropriate fewer turn number and surface area suitable for higher frequency range for the study of whistler phenomenon.

Inappropriate general statements

1. p.1, The abstract states that geospace activities critically affect advanced technologies as well as climate change is to me inappropriate as climate change as nothing to do with this study so why mentioning it?

Thank you for this comment. Because this section provides a perspective within the broader context of a significance on the research trend, it has no direct connection to the scope of the

present study, as pointed out by the reviewer. Upon preparing this statement, we followed Nature Communications' Abstract Guidelines, which recommend indicating "implications". We understand that the degree of implications is debatable, but we still believe that it is valuable to provide a broader context and potential future applicability of the present research topics to a reader. As far as our knowledge goes, in the long-term perspective of space weather and geospace research, there is a relatively widely-understood recognition that the effects of space weather activities affects the long-term climate change on Earth, as well as phenomena such as aurora occurrences, satellite malfunctions, disruptions power grids, etc.. Although our study does not at present directly useful for the studies of such problems, as pointed out, we believe that progress of the understanding of the mechanism of wave-particle interaction in plasma is essential towards the better understanding of the diverse space weather phenomena. On this basis, we decided to revise these perspective words so that overstatement and misunderstanding are avoided, and move them to the concluding remarks so that the present work and the future perspective is clearly distinguished.

Now the abstract ends with the following rather simple explanation.

We anticipate that the laboratory realisation of chorus-like whistler waves in magnetospheric geometry will accelerate the synergistic investigation of space weather systems.

In pages 12 and 13, we revised the concluding remark as follows.

In this work, spontaneously excited chirping electromagnetic and rather broadband electrostatic waves were experimentally investigated in the RT-1 levitated dipole. Chorus emission has been shown to appear ubiquitously in laboratory and space environments for various parameters when plasma satisfies certain conditions. Investigations conducted in a controlled laboratory environment with higher reproducibility and high-resolution diagnostics are expected to provide a new complementary approach to further understanding the chorus emission phenomena that are common in both laboratory and geospace plasmas. In the future, this research might contribute to the understanding of wave particle interactions in the space weather system, which has a critical impact on advanced technologies and climate change [1].

2. p.2, the importance of space weather phenomena has necessitated active research in this area. This sentence is not necessary as chorus has been actively studied both observationally and theoretically since their discovery and particularly for the past 20 years, so it is not space weather which has necessitated active research in this area.

We appreciate this comment. We completely agree with the reviewer's comment that "necessitated" is not an appropriate wording in the context of long history of study in this area. We revised this sentence as follows, in page 2 (here "this area" means chorus):

Active research in this area has offered some important insights into the space weather phenomena.

Minor

Thank you very much for the careful and kind suggestions to improve the manuscript. We revised these points accordingly.

p.3, RT-1 (Ring Trap-1) shall be defined the first time it is mentioned on p.3 and not on p.4

We corrected the sentence in page 3 as follows.

In this study, we focus on the higher-frequency fluctuation activity of Ring Trap-1 (RT-1) and report the ...

p.5, reference to Figure 2b shall be Figure 2c

P.5, reference to Figure 2c shall be Figure 2d

We corrected them accordingly in page 5.

Minor typos

p.2 please correct to proper English the following sentence: helping the understand

We modified this sentence to "helping the understanding of" ... in page 2.

p.4, caption of Figure 1, first word shall appear as "Inside" not "Inis de"

This was corrected to "Inside" in page 4.

p.5, please correct "on the order"

This phrase was modified to "around 10 eV" in page 5.

p.10, please replace "geospacer" by "geospace"

This was corrected to "geospace".

Thank you very much again for the careful and helpful review on the manuscript.

Replies to Reviewer #3:

Thank you very much for the positive comments, and for the insightful suggestions that helped us to greatly improve the manuscript. We studied the raised comments and revised the manuscript accordingly. In the following, we write the reviewer comments in red, our replies in blue, and text in the manuscript in green.

This paper summarizes experiments on the levitated dipole experiment RT-1 where spontaneous generation of chirping whistler waves are observed. The paper identifies the free energy source as temperature anisotropy produced through electron cyclotron resonance heating. The paper reports on the observations of the frequency and wave-vector measurements and their toroidal coherence in the device. The paper is well written and very exciting scientific progress. I recommend publication. I would encourage the authors to make two minor additions to the paper to help put this experiment in the proper context.

We deeply appreciate your kind words on our work, and for the valuable suggestions.

First, laboratory studies of chorus like phenomenon were also performed at the US Naval Research Laboratory in a linear device (e.g. [1]) where both spontaneous emission and stimulated emission (or triggered) chorus was studied.

References

[1] E. M. Tejero, C. Crabtree, D. D. Blackwell, W. E. Amatucci, G. Ganguli, and L. Rudakov, Experimental characterization of nonlinear processes of whistler branch waves. *Physics of Plasmas*, 23(5):055707, 2016.

We appreciate the reviewer for raising our awareness about this important previous study. We added this publication in the citation list as [21] and placed it in an appropriate context of the main text, in page 2.

Second, in figure 4 the measured wavelength was reported as between 1 and 0.3 meters which is comparable to the size of the plasma (also reported in Figure 4). This has strong implications for the generation mechanism here since the wavelength is comparable to the system gradient scale size. This is in contrast to previous laboratory experiments (at UCLA and NRL) and in contrast to the Earth's radiation belts where WKB approximations are more appropriate.

Thank you for the comment. We appreciate your insightful pointing out and believe that such difference is important also for the further understanding of chorus emission and related wave particle interaction phenomena. In Page 11, we added the following explanation.

Despite being whistler waves caused by the temperature anisotropy of high energy electrons, it is noteworthy that the wavelengths observed differ significantly due to differences in the equipment and environment [19-23]. Further studies are needed for the comparison of the wave excitation and propagation properties with other experiments.

We also added the following sentence in the concluding remark in page 12.

Chorus emission has been shown to appear ubiquitously in laboratory and space environments for various parameters when plasma satisfies certain conditions.

Also, in relation to the dispersion relation of the wave, we have conducted additional measurements of the dispersion relation as shown in Fig.5d in page 10.

Fig.5. d, Dispersion relation for local electron density and magnetic field strength at the Bdot probe position (line) and measured values of wave number and frequency (circles).

Thank you very much again for the instructive and helpful review on our manuscript.

REVIEWERS' COMMENTS

Reviewer #1 (Remarks to the Author):

The manuscript has been significantly improved. The Authors did a great work and answered all the comments. There are a few editorial comments and typos, which I think could be eliminated during the proofreading process. I think that the paper could be accepted without any further comments.

Reviewer #2 (Remarks to the Author):

Title

Definitely more appropriate

Abstract

Please correct was follows

In a hot electron high- β plasma (β is the ratio of the plasma pressure to the magnetic pressure)

Question mark

We anticipate that the laboratory realisation of chorus-like whistler waves in magnetospheric geometry will accelerate the synergistic investigation of space weather systems.

What do you mean by accelerate the synergistic investigation of space weather systems?

Space weather systems is definitely not an expression commonly used the field of space weather.

Your study may contribute to better understand the generation of chorus waves. These waves do play a role in physical processes like particles precipitation that is related to space weather impacts. But the expression above is barely understandable. Please rephrase it.

Core of the manuscript

Thanks for taking into account all my remarks.

Conclusion

Disagreement (last sentence)

In the future, this research might contribute to the understanding of wave particle interactions in the space weather system, which has a critical impact on advanced technologies and climate change

There is no such thing as the space weather system. Space Weather describes the phenomena that impact systems and technologies in orbit and on Earth. Mentioning climate change here is misleading. Climate change is due to human activity and shall not be linked to space weather, i.e. ultimately the Sun activity. Climate change shall simply not be mentioned in this context.

Reviewer #3 (Remarks to the Author):

The authors have substantially improved the manuscript with many important details added that were obtained by further laboratory experiments and new analysis techniques. In my opinion, this paper represents substantial and exciting scientific progress. There are many important results that will help guide theory, simulation, and in situ data analysis of the chorus phenomenon. In particular, the role of inhomogeneity in the nonlinear dynamics is emphasized in this levitated dipole experiment that has not been investigated in the laboratory before. For example, the state diagram produced in Figure 6 is a critical element that will challenge theory and modeling for some time. The spatial extent of the chorus emission and the fact that it is toroidally localized is also a critical piece of data. I hope the authors will find a way to obtain more data on the spatial coherence scale in the near future. In short, this manuscript is more than worthy of prompt publication in its present form.

Dear Reviewers,

We deeply appreciate for your time and work for reviewing our manuscript. Here we send the point-by-point response to the comments and revised manuscript.

In the reply letter to the reviewers, we write the **reviewer comments in red**, our replies in blue, and **text in the manuscript in green**. In the manuscript, modifications made are written in blue.

REVIEWERS' COMMENTS

Reviewer #1 (Remarks to the Author):

The manuscript has been significantly improved. The Authors did a great work and answered all the comments. There are a few editorial comments and typos, which I think could be eliminated during the proofreading process. I think that the paper could be accepted without any further comments.

We thank the Reviewer for giving us the positive comment. We also would like to thank again for the careful reading and constructive suggestions that substantially helped us to improve the manuscript.

Reviewer #2 (Remarks to the Author):

We appreciate the comprehensive reading and careful suggestions on the manuscript.

Title

Definitely more appropriate

Thank you for this comment.

Abstract

Please correct was follows

In a hot electron high- β plasma (β is the ratio of the plasma pressure to the magnetic pressure)

We agree to use a plain expression on this point, and modified this phrase accordingly:
high- β (β is plasma pressure normalised by magnetic pressure) plasma

was modified to

high- β plasma (β is the ratio of the plasma pressure to the magnetic pressure)
in the Abstract, in page 1.

Question mark

We anticipate that the laboratory realisation of chorus-like whistler waves in magnetospheric geometry will accelerate the synergistic investigation of space weather systems.

What do you mean by accelerate the synergistic investigation of space weather systems?

Space weather systems is definitely not an expression commonly used the field of space weather. Your study may contribute to better understand the generation of chorus waves. These waves do play a role in physical processes like particles precipitation that is related to space weather impacts. But the expression above is barely understandable. Please rephrase it.

We modified this phrase as follows according to the suggestion.

We anticipate that these experiments will accelerate the laboratory investigation of space weather systems.

We believe that the concern raised by the Reviewer here is now untangled.

Core of the manuscript

Thanks for taking into account all my remarks.

We appreciate your helpful suggestions.

Conclusion

Disagreement (last sentence)

In the future, this research might contribute to the understanding of wave particle interactions in the space weather system, which has a critical impact on advanced technologies and climate change

There is no such thing as the space weather system. Space Weather describes the phenomena that impact systems and technologies in orbit and on Earth. Mentioning climate change here is misleading. Climate change is due to human activity and shall not be linked to space weather, I.e. ultimately the Sun activity. Climate change shall simply not be mentioned in this context.

We deleted the mention on climate according to the concern. Now this sentence is modified to

In the future, this research might contribute to the understanding of wave particle interactions in the space weather system, which has a critical impact on advanced technologies [1].

In page 13, Conclusion,

Thank you again for the valuable comments and suggestions, and we hope that the manuscript is now revised appropriately.

Reviewer #3 (Remarks to the Author):

The authors have substantially improved the manuscript with many important details added that were obtained by further laboratory experiments and new analysis techniques. In my opinion, this paper represents substantial and exciting scientific progress. There are many important results that will help guide theory, simulation, and in situ data analysis of the chorus phenomenon. In particular, the role of inhomogeneity in the nonlinear dynamics is emphasized in this levitated dipole experiment that has not been investigated in the laboratory before. For example, the state diagram produced in Figure 6 is a critical element that will challenge theory and modeling for some time. The spatial extent of the chorus emission and the fact that it is toroidally localized is also a critical piece of data. I hope the authors will find a way to obtain more data on the spatial coherence scale in the near future. In short, this manuscript is more than worthy of prompt publication in its present form.

We are grateful to the reviewer for recognizing the importance of our study and for writing a concise and nice summary of the manuscript. We also appreciate the suggestion on the future work, and will seriously work on this topic in the near future. Thank you again for your constructive comments and helpful suggestions.